# AudiFair: Privacy-Preserving Framework for Auditing Fairness

## Abstract

Ensuring fairness in AI is challenging, especially when privacy concerns prevent access to proprietary models and training data. We propose a cryptographic framework for auditing fairness without requiring model disclosure.

Unlike existing solutions—which either do not capture attack vectors enabling dishonest model providers to manipulate a dataset to pass audits unfairly, or require involving real-world model users to protect against dishonest behaviors—our framework realizes the following properties simultaneously for the first time: 1) **Model Privacy**: Proprietary model details remain hidden from verifiers. 2) **Dishonest Provider Robustness**: Even if model providers are dishonest, a verifier can statistically attest to the fairness of the model without involving real-world users. 3) **Test Data Transparency**: Test data for auditing is generated in a transparent and accountable way, preventing dishonest parties from manipulating it.

We achieve these goals by carefully orchestrating cryptographic commitments, coin tossing, and zero knowledge proofs, and we report the empirical performance for auditing private decision tree models. Our solution is highly communication-efficient, delivering a significant improvement (~200,000x for a 30k-sized dataset) over the current state-of-the-art methods.

## 1 Introduction

As AI adoption grows, organizations face increasing pressure to ensure fairness in machine learning systems (Fuster et al., 2022; ProPublica, 2016; Reuters, 2018). Demonstrating compliance can be difficult, especially when models are proprietary. External examiners often question companies about their training and fairness measures, but this is resource-intensive and yields limited insight. A common alternative is *black-box auditing* (Panigutti et al., 2021; Krafft et al., 2024), where examiners evaluate fairness through API access to the model's public inference interface. For instance, if a proprietary model performs a classification task, the examiner may query the model with sample input data and analyze the outcomes conditioned on sensitive attributes. While black-box auditing preserves the confidentiality of the model, it is vulnerable to model swapping, and a dishonest model provider could return fabricated outputs just for the sake of passing the audit.

Thus, what is needed is the ability to certify desired properties, e.g., fairness above a given threshold, while keeping model details such as weights private and ensuring that the same model is used both during audit and deployment. Recent frameworks for such *privacy-preserving* and *certifiable* ML use cryptographic and differential privacy techniques. Several works target fairness certification in particular, but existing approaches still trade off security, the scope of fairness metrics, or practicality.

Among them, only two works—FairProof (Yadav et al., 2024) and OATH (Franzese et al., 2024)—provide security against *dishonest model providers* (see Table 1), deviating from the protocol specifications. Both, however, require client participation in fairness verification, which is problematic since it exposes real users to potentially unfair models with harmful consequences. OATH mitigates this with a pre-deployment certification step between provider and auditor, but depending on the implementation, a dishonest provider may circumvent this check when deploying the model (see Section 2). Moreover, neither work considers malicious auditors who may falsely accuse an honest provider—e.g., to benefit a competitor who bribed them.[1]

---

[1] This risk is less relevant in Yadav et al. (2024), where there is no audit per se; instead, clients can request certificates for (individual) fairness guarantees on their own queries.

Table 1: Properties and security guarantees of privacy-preserving works on fairness certification. ✓= supported; ✗= not supported; ✓= required; ✗= not required.

| Work | Model | Fairness Type | Dishonest Provider Robustness | Test Data Transparency | Continuous verification |
|------|-------|---------------|-------------------------------|------------------------|-------------------------|
| Shamsabadi et al. (2023) | DT/RF | Group | ✗ | ✗ | ✗ |
| Yadav et al. (2024) | NN | Individual | ✓ | ✗ | ✓ |
| Franzese et al. (2024) | General | Group | ✓ | ✗ | ✓ |
| Bourrée et al. (2025) | General | Group | ✗ | ✗ | ✗ |
| **This work** | DT/RF | Group | ✓ | ✓ | ✗ |

Continuous verification requires model provider's clients to participate in verification during deployment.

All in all, it is crucial to design an auditing framework that does not rely on additional parties such as clients, and is secure under the following threat models: (1) **Dishonest Model Providers:** A provider might deviate from the protocol specifications to pass the fairness check with an unfair model, and (2) **Dishonest Model Verifiers:** A verifier might deviate from the protocol specifications to break the privacy of the model or the training data, or to misrepresent the fairness of the model.

We overcome the challenges mentioned above by proposing AudiFair, a privacy-preserving framework for certifying model fairness that uses cryptographic commitments and zero-knowledge proofs. AudiFair prevents potentially dishonest providers from bypassing the fairness checks by requiring them to prove the fairness using a *synthetically generated* fresh test dataset, and does not require involving real-world clients in the auditing process. Along the way, we identify the following key properties that a privacy-preserving auditing framework should satisfy, and formally prove that AudiFair satisfies them:

**Model Privacy**: Model providers can protect the privacy of their proprietary model parameters and training data from any external verifiers.

**Dishonest Provider Robustness**: Even if model providers are dishonest, a verifier can statistically attest to the fairness of the proprietary model, using a jointly generated synthetic test dataset and well-established fairness metrics.

**Test Data Transparency**: Neither verifiers nor model providers can arbitrarily deviate from the prescribed test data generation process. This ensures that the test data used for fairness auditing cannot be manipulated to artificially influence the outcome.

Our fairness certification framework is model-agnostic, but for implementation and evaluation we focus on AudiFair instantiated with decision trees trained on standard benchmark datasets. We significantly improve upon C-PROFITT in terms of communication bandwidth, achieving ∼200,000x improvement for a dataset with 30,000 datapoints and 23 features, albeit at the cost of a ∼10x higher prover time and a one-time setup phase.

## 2 RELATED WORK

The most related to us are OATH and C-PROFITT, which utilize zero-knowledge proofs to certify (group) fairness. In terms of security, in contrast to us, C-PROFITT does not guarantee fairness if the model provider is *dishonest* and deviates from the protocol specifications. This is because C-PROFITT lets the model provider select the dataset using which fairness is certified, and assumes—rather than enforces—that it matches the training data. As a result, a dishonest provider can pass fairness checks with an arbitrary "unfair" model by choosing a maliciously crafted dataset. While this limitation was recently pointed out in OATH (Franzese et al., 2024), we show an even easier attack in App. D.

To mitigate the risk of dishonest model providers manipulating data, OATH introduces additional parties who supply independent samples. It involves three roles: the client, who provides a data point for classification; the prover (i.e., model provider); and the verifier, who assesses fairness with respect to both the calibration dataset and client queries. The calibration dataset may be selected either by the prover or by the verifier, depending on the implementation. However, when chosen by

the prover, the fairness check is again vulnerable to data-forging attacks—same as C-PROFITT. In such cases, OATH falls back to guarantees based solely on the client's queries. However, another problem is that a malicious client may itself launch a similar attack by choosing their queries in an adversarial way. This is not excluded by the model, as OATH specifically aims to protect against situations where a malicious client, frustrated by not achieving a desired outcome, seeks to retaliate by framing the prover negatively in the eyes of the verifier. Our test data transparency helps with mitigating this issue.

Further, similar to our work, recent studies (Bourrée et al., 2025; Yuan & Wang, 2025) investigate the use of synthetic data for fairness auditing. In contrast to us, however, these works do not provide cryptographic guarantees – in particular, Bourrée et al. (2025) is vulnerable to model swapping attacks, where the model provider swaps the model during the audit in order to pass the audit, and Yuan & Wang (2025) assumes a fully trusted auditor and lets model provider send the model to this auditor in clear.

Finally, Zhang et al. (2020) introduced zk proofs for inference and accuracy of decision trees. This work serves as a basis for the instantiation of our framework when the model is a decision tree. In contrast to our work, Zhang et al. (2020) target inference and accuracy rather than fairness and do not provide test data transparency.

We discuss further related works in Appendix C.

## 3 PRELIMINARIES AND SETTING

**Notation** We use a bracket notation $[n]$ to denote a set of integers $\{1, \ldots, n\}$. We denote the decision tree by $\mathcal{T}$, the height of the tree by $h$, and the number of attributes by $d$, respectively. A datapoint $\mathbf{a}$ is represented by a key-value table, i.e., $\mathbf{a} = \{\mathsf{attr}_1 : \mathsf{val}_1, \ldots, \mathsf{attr}_d : \mathsf{val}_d\}$. The sensitive attribute is denoted by $s \in [d]$. We denote a cryptographic hash function by $\mathsf{H}()$. The number of test data points is $n$. For any algorithm $\mathcal{A}$, we denote its output on input $x$ by $y \leftarrow \mathcal{A}(x)$. Whenever the randomness $\rho$ is sampled by $\mathcal{A}$ internally, we write $y \xleftarrow{\rho} \mathcal{A}(x)$. We denote the process of uniformly sampling an element $x$ from a set $S$ by $x \xleftarrow{\$} S$. We denote the function $f(\lambda)$ is *negligible* if for every polynomial $p(\lambda)$, there exists $\lambda_0$ such that for all $\lambda \geq \lambda_0$, $f(\lambda) < \frac{1}{p(\lambda)}$. A function $f(\lambda)$ is said to be *overwhelming* if $1 - f(\lambda)$ is negligible.

**Fairness** Over the years, numerous fairness metrics have emerged Heidari et al. (2019), each grounded in distinct philosophical and societal perspectives. These notions are often conflicting and there is no consensus on a single, "universal" definition of fairness that suits all use cases. We emphasize that it is not our goal to identify the most appropriate notion, nor is it to advertise for any specific definition. We focus on the following traditional group-fairness notions:

- **Demographic Parity (DP)** ignores the ground truth and aims to equalize the probability of a positive classifier output in each sensitive group. Let $\hat{Y}$ denote the predicted outcome, then:
$$\mathrm{DP}(\hat{Y}; s) = \left| \Pr\left[\hat{Y} = 1 \mid s = 0\right] - \Pr\left[\hat{Y} = 1 \mid s = 1\right] \right|$$

- **Equalized Odds (EqOd)** aims to equalize the false positive and true positive rates in each sensitive group. Let $\hat{Y}$ denote the predicted outcome, and $Y$ denote the true outcome. Then:
$$\mathrm{EOd}(\hat{Y}; s) = \max_{y \in \{0,1\}} \left| \Pr\left[\hat{Y} = 1 \mid s = 0, Y = y\right] \right.$$
$$\left. - \Pr\left[\hat{Y} = 1 \mid s = 1, Y = y\right] \right|$$

We also consider the MRD metric, which is used by major US banks to evaluate the fairness of credit scoring models:[2]

- **Mean Residual Difference (MRD)** measures systematic bias in the model's prediction errors (Zink & Rose, 2020; Corbett-Davies et al., 2023). If one group consistently has higher or lower residuals, this suggests unfair treatment.
$$\mathrm{MRD}(\hat{Y}; s) = \left| \mathrm{E}[Y - \hat{Y} \mid s = 0] - \mathrm{E}[Y - \hat{Y} \mid s = 1] \right|$$

---
[2]Private communication

**Cryptographic Commitment** Cryptographic ML certification often uses *commitment schemes* (Blum, 1981). In particular, commitments allow a model provider to bind itself to a model in a way that hides model internals, yet prevents the provider from swapping models during or after certification. Formally, a commitment scheme COM is a tuple of the following algorithms:

- A *commitment algorithm* com $\overset{\rho}{\leftarrow}$ Commit(msg): Takes as input a message msg $\in \{0,1\}^{\ell_m}$, internally samples a randomness $\rho \in \{0,1\}^{\ell_r(\lambda)}$, and returns a commitment com $\in \{0,1\}^{\ell_c(\lambda)}$. Here $\ell_m, \ell_r, \ell_c$ are some polynomials in $\lambda$, the security parameter[3].

- An *opening algorithm* $b \leftarrow$ Open(com, msg, $\rho$): Outputs a decision bit $b \in \{0,1\}$ indicating whether an opening of the commitment is valid or not.

We require standard security properties: *Hiding* (i.e., com reveals nothing about the committed message $m$) and *binding* (i.e., the sender cannot open com to two distinct messages $m$ and $m'$). See Appendix A.3 for a formal treatment. In the instantiation of our framework for decision trees, we rely on a commitment scheme optimized for decision trees; see Appendix E.1 for more.

**Non-Interactive Zero-Knowledge Proofs** Another key technique in cryptographic ML certification are the *zero-knowledge proofs*. A zero-knowledge proof is a cryptographic protocol between two parties that allows the *prover* to convince the *verifier* that a *statement* is true *without revealing any information beyond its validity*. The statement's validity is formally defined by an NP relation $\mathcal{R}$: a statement x is valid if there exists a *witness* w such that $(x, w) \in \mathcal{R}$. The statement is public and known to both parties, while the witness is private and only known to the prover. For example, one can prove that a model $h$ outputs $y$ on a public input $a$ *without revealing anything about $h$'s weights*. Here, the *statement* is $x = (a, y)$, and the witness is $w = h$.

Formally, a non-interactive zero knowledge proof system (henceforth NIZK) is defined for an NP relation $\mathcal{R}$, i.e., a set of *public statements* x and *private witnesses* w, where the size of w is bounded by a polynomial in the size of x, and $(x, w) \in \mathcal{R}$ can be checked in polynomial time given $(x, w)$. It is denoted by a tuple ZK = (Gen, Prove, Verify) of three algorithms:

- pp $\leftarrow$ Gen($1^\lambda$) is a setup algorithm that samples a public parameter pp, where $\lambda$ denotes a security parameter.

- $\pi \leftarrow$ Prove(pp, x, w) is a prover that outputs a proof $\pi$ asserting $(x, w) \in \mathcal{R}$. If $(x, w) \notin \mathcal{R}$, Prove outputs $\perp$.

- $b \leftarrow$ Verify(pp, x, $\pi$) is a verifier that outputs a decision bit $b \in \{0, 1\}^*$.

We assume ZK to satisfy standard security properties: *completeness* (i.e., if Prove and Verify follow the protocol, Verify always accepts), *(knowledge) soundness* (i.e., if Verify accepts the proof generated by a cheating prover $\mathcal{A}$, then it must be that $\mathcal{A}$ owns a valid witness w satisfying given NP relation w.r.t. statement x), and *zero knowledge* (i.e., if the proof generated by Prove leaks nothing except that $\exists$w such that $(x, w) \in \mathcal{R}$). See Appendix A for a formal treatment.

**Problem Setting** We consider a setting where a model provider holds a proprietary classification model that must remain private. An auditor seeks to verify fairness properties of this model—for example, whether its demographic parity gap is below a specified threshold on a uniformly random sample from the target distribution. The model provider may be malicious and attempt to pass the audit with a model that *does not* satisfy the required fairness guarantee. Conversely, the auditor may be malicious and (i) attempt to extract proprietary information about the model or (ii) misrepresent the achieved fairness guarantee, causing the Prover to fail the audit despite holding a fair model. In the following, we will refer to the model provider as *Prover*, and to the auditor as *Verifier*.

## 4 ABSTRACT FAIRNESS AUDITING FRAMEWORK

We now design a mechanism which allows to prove that a proprietary model satisfies certain fairness criteria without revealing anything about the model itself. Our framework applies to arbitrary

---

[3]Informally, the security parameter $\lambda$ sets the computational hardness of breaking the scheme; e.g., with $\lambda$=128 any attack should need about $2^{128}$ work, which is considered infeasible.

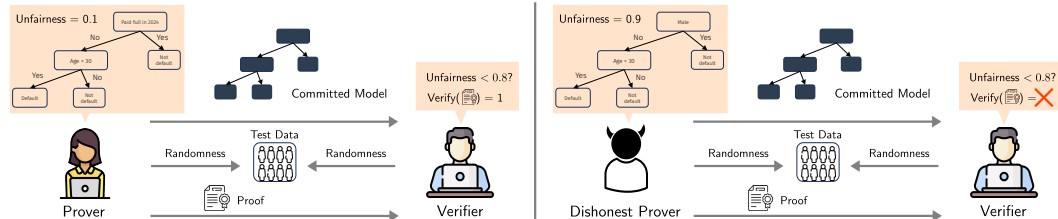

Figure 1: Overview of our framework AudiFair for auditing fairness of proprietary models. **Left**: An honest prover computes the fairness outcome and convinces the verifier. **Right**: A dishonest prover with the model that does not pass the fairness check attempts to cheat during the proof generation, but the verifier rejects. Icons are retrieved from flaticon.com.

classification models. We first present a high-level overview of our protocol (Section 4.1), provide a formal definition of the auditing framework and its security properties (Section 4.2), and then present a generic construction of our protocol (Section 4.3).

## 4.1 OVERVIEW

The key idea in our framework it to check fairness on a *freshly sampled synthetic test dataset*. While recently, a number of works (Bourrée et al., 2025; Yuan & Wang, 2025) proposed to utilize synthetic data generation for ML fairness auditing (see Section 2), these works assume that the model provider (Bourrée et al., 2025) or the auditor Yuan & Wang (2025) are trusted. We now explore this idea in the context of *cryptographic* ML certification that is secure even if the corresponding parties misbehave. We further empirically verify the effectiveness of testing fairness on synthetic data in Sec. 5. Our framework is comprised of three phases:

(a) **Commit**: Model Provider (or Prover) $\mathcal{P}$ generates a commitment com to the model $M$, and sends com to Model Verifier $\mathcal{V}$.

(b) **Data Generation**: $\mathcal{P}$ and $\mathcal{V}$ jointly compute and agree on the randomness $r$ for generating the synthetic test dataset $\mathcal{D}$ and the corresponding ground truth labels $\mathcal{Y}$, according to a prescribed synthetic data generation algorithm DataGen.

(c) **Prove Fairness**: Using the commitment com, the test dataset $\mathcal{D}$, and the ground truth labels $\mathcal{Y}$, $\mathcal{P}$ and $\mathcal{V}$ conduct zero-knowledge proof that committed $M$ satisfies the fairness condition with respect to $\mathcal{D}$ and $\mathcal{Y}$.

In more detail, in Step **(a)**, the Prover sends a cryptographic commitment com to the model, which binds the prover to the specific model $M$ (while hiding its internals). Crucially, we require that the Prover does so without knowing the exact dataset $\mathcal{D}$ and the corresponding ground truth labels $\mathcal{Y}$ that will be used for the fairness check: If the order is reversed, i.e., a dishonest Prover first receives $\mathcal{D}$ and $\mathcal{Y}$, then it can commit to a model that is overfitted to the exact test dataset, thus cheating the fairness check.

In Step **(b)**, after the Prover and the Verifier jointly sample a uniformly random seed $r$, they locally derive $(\mathcal{D}, \mathcal{Y}) \leftarrow \text{DataGen}(r)$. This step prevents either party from manipulating the data generation process to their advantage. In our concrete instantiation presented later, we realize this step by interleaving Blum's two-party coin tossing protocol (Blum, 1981). Alternatively, one could rely on a secure randomness beacon (e.g., of Standards & Technology (2018); Baum et al. (2023)) to retrieve fresh and uniform randomness from an external source.

Finally, Step **(c)** can be further decomposed into the following two-stage operations:

**Computation of Fairness Score** Prover first computes the prediction for the given public test dataset $\mathcal{D} = \{\mathbf{a}_1, \ldots, \mathbf{a}_n\}$, by evaluating a previously committed model $M$ on each data point $\mathbf{a}_i$. Finally, the prover uses the predicted labels $\hat{\mathcal{Y}} = \{\hat{y}_i\}_{i \in [n]}$ (along with the test data and the corresponding ground-truth $\mathcal{Y}$) to obtain the fairness result. We abstract the process of calculating fairness metric by a function $\text{FM}(\mathcal{D}, \mathcal{Y}, \hat{\mathcal{Y}}, s)$, where $s$ is a binary sensitive attribute. The function FM can be in-

stantiated in different ways, depending on which fairness metric is employed from Section 3. For instance, to calculate Demographic Parity, one may define

$$\mathsf{FM}(\mathcal{D}, \mathcal{Y}, \hat{\mathcal{Y}}, s) := \left| \frac{|\{\mathbf{a}_i \in \mathcal{D} \ : \ \mathbf{a}_i[s] = 0, \hat{y}_i = 1\}|}{|\{\mathbf{a}_i \in \mathcal{D} \ : \ \mathbf{a}_i[s] = 0\}|} - \frac{|\{\mathbf{a}_i \in \mathcal{D} \ : \ \mathbf{a}_i[s] = 1, \hat{y}_i = 1\}|}{|\{\mathbf{a}_i \in \mathcal{D} \ : \ \mathbf{a}_i[s] = 1\}|} \right| \quad (1)$$

and FM can be instantiated for other metrics analogously. Prover then checks that $\mathsf{FM}(\mathcal{D}, \mathcal{Y}, \hat{\mathcal{Y}}, s) \leq t$ with respect to a pre-agreed threshold $t$ between the Prover and the Verifier.

**Zero-knowledge Fairness Proof** Assuming that the fairness check passes, in the last step, the Prover uses a secure NIZK scheme on the previously computed data (predicted labels and fairness outcome) along with the synthetic testdata $\mathcal{D}$ and commitment com to show the following statement in zero-knowledge: I know a model $M$ for which all of the following holds:

(1) It is committed under com

(2) $M$'s prediction for $\mathbf{a}_i$ corresponds to $\hat{y}_i$ for each $i \in [n]$

(3) When the agreed upon fairness metric function FM is computed on the tuple $(\mathcal{D}, \mathcal{Y}, \hat{\mathcal{Y}} = \{\hat{y}\}_{i \in [n]}, s)$, the outcome is below threshold $t$.

Note that our protocol allows for a flexible choice of the ZK scheme to prove these conditions. Further, note that because we use a secure ZK, the protocol does not leak predictions $\hat{\mathcal{Y}}$ w.r.t test data $\mathcal{D}$; Prover only leaks the fact that a committed model $M$ meets the fairness condition w.r.t. a given test data set, sensitive attribute, and fairness threshold.

**Extension 1: Proving exact fairness scores** Above, we focused on the case where the Prover hides the exact outcome of FM and the Verifier only learns a binary outcome (i.e., whether the committed model is fair or not with respect to a public threshold $t$). If the Verifier is interested in understanding the strengths or failings of the committed model, then one can easily tweak our protocol by having the Prover reveal a fairness score $v$ and generate a proof that $v = \mathsf{FM}(\mathcal{D}, \mathcal{Y}, \hat{\mathcal{Y}}, s)$ holds in Step (3). As this merely skips a comparison operation, the performance of this variant is the same as the original protocol.

**Extension 2: Supporting non-binary sensitive attributes** While we focus on binary sensitive attributes for simplicity, it is possible to extend our protocol to non-binary attributes. For this, we compute the fairness score by simply comparing *each* protected minority group to the majority group, and taking the maximum of the (absolute) differences.

### 4.2 DEFINITION OF PRIVACY-PRESERVING FAIRNESS AUDITING

We formally define the syntax and the security properties of the auditing framework sketched above. While zero-knowledge proofs alone (Section 3) do not specify *how* and *when* a public statement x is generated, our tailored syntax and security notions precisely define the requirements for generating x—consisting of a commitment to the model and the synthetic data—in a specific way to prevent manipulation of the fairness metric.

**Definition 1.** *Define an NP relation*
$$\mathcal{R}_{\mathsf{fair}} = \left\{ ((\mathcal{D}, \mathcal{Y}), M) \ : \ \hat{\mathcal{Y}} = M(\mathcal{D}) \wedge \mathsf{FM}(\mathcal{D}, \mathcal{Y}, \hat{\mathcal{Y}}, s) \leq t \right\}$$
*for fairness auditing parameterized by a sensitive attribute $s$, fairnss threshold $t$, and fairness metric* FM. *Let* DataGen *be a randomized synthetic data generation algorithm with $k$-bit randomness space. Let* COM $=$ (Commit, Open) *be a commitment scheme that supports committing to a model $M$ using randomness $\rho$. Let* Setup$(1^\lambda)$ *be a setup algorithm that generates public parameters* pp. *A* privacy-preserving fairness auditing protocol $\Pi$ *for* $\mathcal{R}_{\mathsf{fair}}$ *and* DataGen *consists of* Setup, COM, *and an interactive process between Model Owner (or Prover) $\mathcal{P}$ and Verifier $\mathcal{V}$.*

*We denote by $((\mathcal{D}_\mathcal{P}, \mathcal{Y}_\mathcal{P}), (\mathcal{D}_\mathcal{V}, \mathcal{Y}_\mathcal{V}, b)) \leftarrow \langle \mathcal{P}(M, \rho), \mathcal{V} \rangle(\mathsf{pp}, \mathsf{com})$ the following process: given $(M, \rho)$ as a private input to $\mathcal{P}$, and* pp *and* com *as common inputs to both parties, $\mathcal{P}$ and $\mathcal{V}$ interact in a way that $\mathcal{P}$ eventually halts by locally outputting a pair $(\mathcal{D}_\mathcal{P}, \mathcal{Y}_\mathcal{P})$ of synthetic data and ground truth labels, while $\mathcal{V}$ halts by locally outputting $(\mathcal{D}_\mathcal{V}, \mathcal{Y}_\mathcal{V})$, and a decision bit $b \in \{0, 1\}$.*

---

**Protocol 1: Generic Construction of $\Pi$**

$\underline{\mathsf{Setup}(1^\lambda)}$: Run $\mathsf{pp} \leftarrow \mathsf{ZK.Gen}(1^\lambda)$. Output $\mathsf{pp}$.

$\underline{\text{Interactive protocol } \langle \mathcal{P}(M, \rho), \mathcal{V} \rangle(\mathsf{pp}, \mathsf{com})}$: $\mathcal{P}$ and $\mathcal{V}$ run the following protocol sequentially. If any check fails, the protocol aborts and $\mathcal{V}$ halts by outputting $b = 0$.

1: $\mathcal{V}$ sends $h_2 = \mathsf{H}(r_2)$, where $r_2 \leftarrow \{0, 1\}^k$
2: $\mathcal{P}$ sends $h_1 = \mathsf{H}(r_1)$, where $r_1 \leftarrow \{0, 1\}^k$
3: $\mathcal{V}$ sends $r_2$
4: $\mathcal{P}$ checks that $h_2 = \mathsf{H}(r_2)$ and does:

    (a) $r \leftarrow r_1 \oplus r_2$
    (b) $(\mathcal{D}_\mathcal{P}, \mathcal{Y}_\mathcal{P}) \leftarrow \mathsf{DataGen}(r)$
    (c) check $\mathsf{FM}(\mathcal{D}_\mathcal{P}, \mathcal{Y}_\mathcal{P}, M(\mathcal{D}_\mathcal{P}), s) \leq t$
    (d) $\pi \leftarrow \mathsf{ZK.Prove}(\mathsf{pp}, (\mathsf{com}, \mathcal{D}_\mathcal{P}, \mathcal{Y}_\mathcal{P}), (M, \rho))$
    (e) send $(\pi, r_1)$ to $\mathcal{V}$
    (f) locally output $(\mathcal{D}_\mathcal{P}, \mathcal{Y}_\mathcal{P})$
5: $\mathcal{V}$ checks that $h_1 = \mathsf{H}(r_1)$ and does:

    (a) $r \leftarrow r_1 \oplus r_2$
    (b) $(\mathcal{D}_\mathcal{V}, \mathcal{Y}_\mathcal{V}) \leftarrow \mathsf{DataGen}(r)$
    (c) run $b \leftarrow \mathsf{ZK.Verify}(\mathsf{pp}, (\mathsf{com}, \mathcal{D}_\mathcal{V}, \mathcal{Y}_\mathcal{V}), \pi)$
    (d) locally output $(\mathcal{D}_\mathcal{V}, \mathcal{Y}_\mathcal{V}, b)$

---

We are ready to state the security properties of $\Pi$. For simplicity, we state the properties informally below, and defer the exact definitions of other properties to Appendix F.

**Completeness** requires that if both the Prover and Verifier honestly follow the protocol, and the Prover owns a fair model $M$, then the Verifier is always convinced *and* they agree on the same synthetic data.

**Binding:** Once the Prover commits to a model, it cannot change it without detection.

**Dishonest Provider Robustness:** The protocol is robust against dishonest model providers, meaning that if $\mathcal{P}$ tries to cheat by committing to an unfair model $M$ (i.e., not satisfying the fairness condition described by $\mathcal{R}_{\mathsf{fair}}$), the $\mathcal{V}$ can detect it by outputting $b = 0$.

**Model Privacy:** The protocol leaks no information about the model $M$ to $\mathcal{V}$ except that it satisfies the fairness condition described by $\mathcal{R}_{\mathsf{fair}}$.

**Test Data Transparency:** The protocol ensures that neither party can bias the distribution of the test dataset $\mathcal{D}$ generated during the interactive protocol, provided the protocol does not abort.

**Remark 1** (Running $\Pi$ Multiple Times). *If $\Pi$ is run multiple times with the same commitment $\mathsf{com}$, it is crucial that $\Pi$ is binding. This is to ensure that the Prover cannot cheat by secretly swapping models between different runs of the protocol to pass the fairness check in case the originally committed model fails. Thanks to the combination of binding and dishonest provider robustness, the Prover can only convince the Verifier with a uniquely determined model that satisfies the fairness condition for all runs of the protocol.*

### 4.3 GENERIC CONSTRUCTION AND SECURITY

We provide a generic construction of privacy-preserving fairness auditing protocol in Protocol 1. Other than COM, the protocol utilizes the following cryptographic building blocks as subroutines: hash function $\mathsf{H} : \{0, 1\}^* \to \{0, 1\}^\ell$, and non-interactive zero-knowledge (NIZK) proof system $\mathsf{ZK} = (\mathsf{Gen}, \mathsf{Prove}, \mathsf{Verify})$ for "commit-and-prove" relation defined as

$$\mathcal{R}_{\mathsf{cp}} := \left\{ ((\mathsf{com}, \mathcal{D}, \mathcal{Y}), (M, \rho)) : \begin{array}{l} \mathsf{Open}(\mathsf{com}, M, \rho) = 1 \\ \wedge ((\mathcal{D}, \mathcal{Y}), M) \in \mathcal{R}_{\mathsf{fair}} \end{array} \right\}$$

Table 2: Performance of our AudiFair protocol for checking Equalized Odds. 'SMP' stands for Security against Malicious Prover. $n$ is the size of the test dataset, $d$ is the number of features, and $h$ is the height of the decision tree. 'Data' includes both computation of $h_1, h_2$ and an execution of DataGen. The running times are in seconds and based on experiments conducted on an Amazon EC2 c7a.12xlarge instance with 96GB RAM. For the first row, we parallelized FFT and elliptic curve operations of the underlying Groth16 NIZK Setup and Prove using 48 vCPUs. For comparison, we restate the benchmarks for C-PROFITT in the rows marked by 'CP'.

| | $(n, d, h)$ | Setup (s) | Data (s) | Prove (s) | Verify (s) | Comm. (kB) | SMP |
|---|---|---|---|---|---|---|---|
| ACSIncome (Ours) | $(15000, 10, 10)$ | 103 | 288 | 52 | $< 1$ | 0.6 | ✓ |
| Credit (CP) | $(30000, 23, 10)$ | – | – | 72 | 72 | 109875 | ✗ |
| Credit (Ours) | $(30000, 23, 10)$ | 1430 | 750 | 826 | $< 1$ | 0.6 | ✓ |
| Adult (CP) | $(45222, 14, 10)$ | – | – | 105 | 105 | 148685 | ✗ |
| Adult (Ours) | $(45222, 14, 10)$ | 1982 | 968 | 1002 | $< 1$ | 0.6 | ✓ |

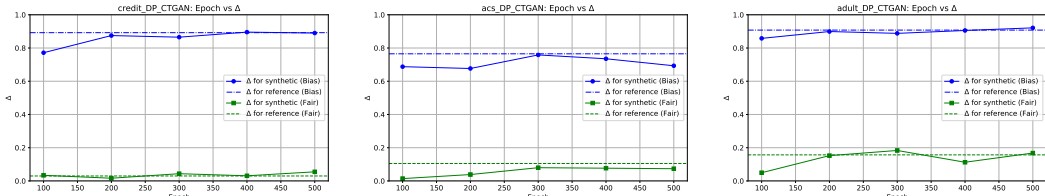

Figure 2: Comparison of Demographic Parity score $\Delta$ calculated using the reference dataset and CTGAN-generated synthetic data with varying epochs (the number of times each data point is used during synthesizer training), with the sensitive attribute set to gender. Dashed lines represent $\Delta$ derived from the reference dataset. Both fair and biased models are trained on 70% of each reference dataset, and the remaining 30% is used for calculating $\Delta$ for the reference set. For the synthetic data, we use the same 30% of the reference dataset to generate synthetic data with CTGAN.

We now state the main security claim of our generic construction in Theorem 1. The formal statement and proof is deferred to Appendix F. Intuitively, dishonest provider robustness is realized by the fact that a dataset is obtained after the Prover commits to a model, and that knowledge soundness of ZK forces the Prover to use a model that was committed earlier and satisfies the fairness condition. Model privacy is achieved by secure instantiation of COM and ZK. Finally, data transparency is achieved by the initial interaction exchanging hashes of randomness shares, which ensures that neither party can bias the randomness used to generate the test dataset.

**Theorem 1 (informal).** *We have the following security properties of Protocol 1: Suppose* ZK *and* COM *satisfy the standard properties defined in App. A.3 and A.4, and the hash function* H *is modeled as a programmable random oracle (App. A.2). Then Protocol 1 has completeness, binding, dishonest provider robustness, model privacy, and test data transparency.*

## 5 CONCRETE INSTANTIATIONS AND EVALUATIONS

### 5.1 AudiFair: INSTANTIATING Π FOR DECISION TREES/RANDOM FORESTS

We now describe how to instantiate our protocol for decision trees (see Appendix B for decision tree background). Each building block of Protocol 1 from the previous section is instantiated as follows; additional implementation details are provided in Appendix E.

- We use SHA3-256, available in hashlib Python library, as our hash function H.

- For our commitment COM we use the Authenticated Decision Tree (ADT) commitment scheme (Zhang et al., 2020) with the SWIFFT hash function (Lyubashevsky et al., 2008) as a subroutine, implemented in C++.

- We use CTGAN (Xu et al., 2019), available in the SDV library (Developers, 2024), as DataGen.

- Finally, we use Groth16 NIZK protocol Groth (2016) as a zero-knowledge proof system ZK, implemented in C++ with low-level routines available in the libsnark library Lab (2014). We instantiated its parameters with the BN254 pairing-friendly elliptic curve. As the most expensive operation of Groth16 consists of Fast Fourier Transform (FFT) and elliptic curve arithmetic, which are parallelizable, we compiled the libsnark library while enabling parallelized execution of these low-level computations. To support $\mathcal{R}_{\mathrm{cp}}$ for decision trees, we instantiate a circuit amenable to Groth16 to prove (1) com = ADT.Commit($\mathcal{T}; \rho$), (2) prediction correctness based on the approach of Zhang et al. (2020), and (3) proof of fairness metric w.r.t. an input data set.

  We note that as Zhang et al. (2020) supports random forest models, our AudiFair instantiation can be extended to support random forests as well.

## 5.2 PERFORMANCE AND EFFECTIVENESS OF FAIRNESS TEST WITH SYNTHETIC DATA

We evaluate fair and biased decision trees trained on three widely-used datasets: ACSIncome Ding et al. (2021), Adult Repository (1996), and Default Credit Repository (2009) by using CTGAN-generated synthetic data. See Fig. 2, which shows that the fairness metric is close to the one with the reference dataset, and thus the fairness test is effective. In Appendix G, we also provide the results for other metrics and synthetic data generation algorithms.

The performance of our AudiFair protocol is summarized in Table 2. AudiFair was executed on the ACS dataset for checking Equalized Odds; performance for other metrics remains essentially unchanged, since proof of correct prediction dominates the running time. For this experiment, we selected 50000 data points from the ACS dataset, out of which $70\%$ (i.e., 35000 entries) was used to train the tree and the remaining $30\%$ (i.e., $n = 15000$ entries) was used to train a synthesizer. Then the synthesizer generated $n$ fresh test samples. For Prove, we further enabled OpenMP with 48 threads in order to showcase our approach is parallelization-friendly. To compare with C-PROFITT, we also report the performance for Equalized Odds using the two largest datasets used by Shamsabadi et al. (2023) without parallelization. Following C-PROFITT, we run ZKP on the entire datasets in this experiment.

As Groth16 ZK only outputs three group elements on BN254 elliptic curves, our AudiFair obtains modest proof sizes. In contrast, the communication bandwidth of C-PROFITT is orders of magnitude larger. Unlike C-PROFITT, however, our AudiFair additionally generates a synthetic test dataset, which is used to prove fairness. In terms of prover running time, our current implementation of AudiFair falls short of C-PROFITT. However, this is mitigated by the fact that, in contrast to C-PROFITT, our AudiFair has provides dishonest provider robustness as well as test data transparency.

## 6 CONCLUSION, LIMITATIONS, AND FUTURE WORK

We proposed a secure solution to prove that a (proprietary and private) decision tree model satisfies given fairness constraints. Our solution achieves model privacy, dishonest provider robustness, and test data transparency simultaneously for the first time. We leave several interesting questions for future work: First, while an abstract framework in Section 4.3 supports arbitrary model types, our concrete instantiation is tailored to decision trees/random forests. Designing efficient protocols to support further models, e.g., the widely used XGBoost Chen & Guestrin (2016), is an important direction. Second, as the prover time mainly hinges on the complexity of the backend NIZK scheme Groth (2016), we leave for future work the improvement of the prover time by employing a more prover-efficient scheme as a backend of our AudiFair. Third, in this work we assume an idealized DataGen that takes as input a randomness and produces a sample from the distribution. It would be interesting to see how well the corresponding distribution represents the real data distribution. Finally, it would be interesting to provide a solution in which the verifier's test data remains *private* while protecting against dishonest behaviors. General-purpose solutions for this scenario such as secure multi-party computation exist, but these are not sufficiently efficient. Obtaining a practical scheme likely requires developing new cryptographic techniques.

ETHICS STATEMENT

This work proposes a cryptographic protocol enabling a model provider to convince a verifier that a proprietary model satisfies certain fairness criteria in a privacy-preserving manner. While we put forward a solution that aims to promote algorithmic fairness, it is important to acknowledge that the definition of fairness can vary across different societal contexts and applications. We emphasize that this study does not endorse any specific fairness metric, but rather focuses on providing a generic tool that can be adapted to various definitions of fairness.

**LLM Use.** This paper has been written with the help of LLM to improve grammar and clarity, and to assist with literature search. To obtain the results summarized in Figures 2, 5, and 6, we have used LLM to help generate the Python code for testing synthetic data generation using CTGAN and TVAE. The authors are responsible for the content of this paper.

REPRODUCIBILITY STATEMENT

We back up our theoretical claims on the security and privacy of our construction (Theorem 1) with formal proofs in Appendix F. We provide a detailed description of our implementation and experimental evaluation in Section 5 and Appendix G.

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

# A  Extended Preliminaries

## A.1  Basic Notation and Definitions in Cryptography

In this section, we include minimal introductory material on the basic notation and definitions used in the paper. For a more comprehensive introduction to cryptography, we refer the reader to standard textbooks such as Katz & Lindell (2020); Goldreich (2001).

For two bit strings of equal length, we denote $a \oplus b$ as the bitwise XOR of $a$ and $b$. For any algorithm $\mathcal{A}$, we denote its output on input $x$ by $y \leftarrow \mathcal{A}(x)$. Whenever the randomness $\rho$ is sampled by $\mathcal{A}$ internally, we write $y \xleftarrow{\rho} \mathcal{A}(x)$. We denote the process of uniformly sampling an element $x$ from a set $S$ by $x \xleftarrow{\$} S$. Let $\lambda \in \mathbb{N}$ be a security parameter. We denote the function $f(\lambda)$ is *negligible* if for every polynomial $p(\lambda)$, there exists $\lambda_0$ such that for all $\lambda \geq \lambda_0$, $f(\lambda) < \frac{1}{p(\lambda)}$. A function $f(\lambda)$ is said to be *overwhelming* if $1 - f(\lambda)$ is negligible.

By PPT, we mean *probabilistic polynomial time*. For any probability distributions $\mathcal{D}_0(\lambda)$ and $\mathcal{D}_1(\lambda)$, we say that $\mathcal{D}_0$ and $\mathcal{D}_1$ are computationally *indistinguishable* if for any PPT algorithm $\mathcal{A}$, the following is negligible in $\lambda$:

$$\left| \Pr\left[1 \leftarrow \mathcal{A}(1^\lambda, \mathcal{D}_0(\lambda))\right] - \Pr\left[1 \leftarrow \mathcal{A}(1^\lambda, \mathcal{D}_1(\lambda))\right] \right|$$

If the above holds for *any* algorithm $\mathcal{A}$ (including unbounded ones), we say that $\mathcal{D}_0$ and $\mathcal{D}_1$ are statistically indistinguishable. If the above quantity is 0, we say that $\mathcal{D}_0$ and $\mathcal{D}_1$ are perfectly indistinguishable.

## A.2  Random Oracle Model

In this work, we assume that the hash function H is modeled as a *random oracle* Bellare & Rogaway (1994). In the random oracle model, H is a function that maps arbitrary-length inputs to uniformly random outputs of a fixed length. In this model, H is assumed to be a public function that is accessible to all parties, including the adversary. Thus, whenever a proof of security is in the random oracle model, it is assumed that a reduction or a simulator can observe queries made by an adversary to the random oracle. The random oracle model is an idealized abstraction of hash functions commonly used in security proofs, allowing for the design of secure and efficient protocols. Importantly, whenever H receives a new query, it responds with a freshly and uniformly sampled value $h \xleftarrow{\$} \{0, 1\}^{\ell(\lambda)}$. Moreover, the random oracle is often assumed to be *programmable*, meaning that a reduction or a simulator can predefine the output of the random oracle for a specific input, as long as the input has not been queried before.

## A.3  Security Properties of Cryptographic Commitments

Our protocol relies on a cryptographic *commitment scheme* Blum (1981). In Appendix Appendix E.1, we define an instantiation of a commitment scheme optimized for decision trees. A commitment scheme COM is a tuple of the following algorithms:

- A *commitment algorithm* $\mathsf{com} \xleftarrow{\rho} \mathsf{Commit}(\mathsf{msg})$: Takes as input a message $\mathsf{msg} \in \{0,1\}^{\ell_m}$ from a message space $\mathcal{M}$, internally samples a randomness $\rho \in \{0,1\}^{\ell_r(\lambda)}$, and returns a commitment $\mathsf{com} \in \{0,1\}^{\ell_c(\lambda)}$. Here $\ell_m, \ell_r, \ell_c$ are some polynomials in $\lambda$, the security parameter.

- An *opening algorithm* $b \leftarrow \mathsf{Open}(\mathsf{com}, \mathsf{msg}, \rho)$: Outputs a decision bit $b \in \{0,1\}$ indicating whether an opening of the commitment is valid or not.

We recall standard security properties of cryptographic commitment schemes.

**Completeness** COM is *complete* if for any message $\mathsf{msg}$ in the message space,

$$\Pr\left[1 \leftarrow \mathsf{Open}(\mathsf{com}, \mathsf{msg}, \rho) \ : \ \mathsf{com} \xleftarrow{\rho} \mathsf{Commit}(\mathsf{msg})\right] = 1$$

**Binding** COM is *binding* if for any PPT adversary $\mathcal{A}$, the following probability is negligible in $\lambda$:

$$\Pr\left[\begin{matrix} M \neq M' & (\mathsf{com}, M, \rho, M', \rho') \leftarrow \mathcal{A}(1^\lambda) \\ \wedge\, b = 1 & : & b \leftarrow \mathsf{Open}(\mathsf{com}, M, \rho) \\ \wedge\, ab' = 1 & & b' \leftarrow \mathsf{Open}(\mathsf{com}, M', \rho') \end{matrix}\right].$$

**Hiding** COM is *hiding* if for any PPT adversary $\mathcal{A}$ and any pair of messages $(\mathsf{msg}, \mathsf{msg}')$ in the message space, the following is negligible in $\lambda$:

$$\varepsilon_{hide} = \left| \begin{matrix} \Pr\left[1 \leftarrow \mathcal{A}(\mathsf{com}) \ : \ \mathsf{com} \leftarrow \mathsf{Commit}(\mathsf{msg}) \quad \right] \\ - \Pr\left[1 \leftarrow \mathcal{A}(\mathsf{com}) \ : \ \mathsf{com} \leftarrow \mathsf{Commit}(\mathsf{msg}') \quad \right] \end{matrix} \right|$$

### A.4 Security Properties of Zero Knowledge Proofs

A (non-interactive) zero knowledge proof system (henceforth NIZK) is defined for *NP relation* $\mathcal{R}$, i.e., a set of *public statements* $\mathsf{x}$ and *private witnesses* $\mathsf{w}$, where the size of $\mathsf{w}$ is bounded by a polynomial in the size of $\mathsf{x}$, and $(\mathsf{x}, \mathsf{w}) \in \mathcal{R}$ can be checked in polynomial time given $(\mathsf{x}, \mathsf{w})$. It is denoted by a tuple $\mathsf{ZK} = (\mathsf{Gen}, \mathsf{Prove}, \mathsf{Verify})$ of three algorithms:

- $\mathsf{pp} \leftarrow \mathsf{Gen}(1^\lambda)$ is a setup algorithm that samples a public parameter $\mathsf{pp}$, where $\lambda$ denotes a security parameter.

- $\pi \leftarrow \mathsf{Prove}(\mathsf{pp}, \mathsf{x}, \mathsf{w})$ is a prover that outputs a proof $\pi$ asserting $(\mathsf{x}, \mathsf{w}) \in \mathcal{R}$. If $(\mathsf{x}, \mathsf{w}) \notin \mathcal{R}$, Prove outputs $\bot$.

- $b \leftarrow \mathsf{Verify}(\mathsf{pp}, \mathsf{x}, \pi)$ is a verifier that outputs a decision bit $b \in \{0,1\}^*$.

The common input $\mathsf{x}$ to both parties is called *statement*, and Prove's private input $\mathsf{w}$ is called *witness*. For instance, if the prover is tasked to prove $\mathcal{T}(a) = y$ for public $a$ and $y$ for a private decision tree $\mathcal{T}$, one can set $\mathsf{x} = (a, y)$ and $\mathsf{w} = \mathcal{T}$.

We recall standard security properties of (non-interactive) zero knowledge proof systems.

**Completeness** ZK is *complete* if for any $(\mathsf{x}, \mathsf{w}) \in \mathcal{R}$,

$$\Pr\left[1 \leftarrow \mathsf{Verify}(\mathsf{pp}, \mathsf{x}, \pi) \ : \ \begin{matrix} \mathsf{pp} \leftarrow \mathsf{Gen}(1^\lambda); \\ \pi \leftarrow \mathsf{Prove}(\mathsf{pp}, \mathsf{x}, \mathsf{w}) \end{matrix}\right] = 1$$

**Knowledge Soundness** ZK is (adaptively) *knowledge sound* if for any PPT adversary $\mathcal{A}$, there exists a (non-blackbox) polynomial time extractor $\mathsf{Ext}_\mathcal{A}$ such that the following probability negligible in $\lambda$:

$$\Pr\left[b = 1 \wedge (\mathsf{x}, \mathsf{w}) \notin \mathcal{R} \; : \; \begin{array}{l} \mathsf{pp} \leftarrow \mathsf{Gen}(1^\lambda); (\mathsf{x}, \pi) \leftarrow \mathcal{A}(\mathsf{pp}) \\ b \leftarrow \mathsf{Verify}(\mathsf{pp}, \mathsf{x}, \pi); \mathsf{w} \leftarrow \mathsf{Ext}_{\mathcal{A}}(\mathsf{pp}) \end{array}\right]$$

**Zero Knowledge** ZK is *zero knowledge* if there exists a PPT simulator $\mathsf{Sim} = (\mathsf{Sim}_1, \mathsf{Sim}_2)$ such that for any PPT adversary $\mathcal{A} = (\mathcal{A}_1, \mathcal{A}_2)$ where $\mathcal{A}_1$ always outputs $(\mathsf{x}, \mathsf{w})$ satisfying $(\mathsf{x}, \mathsf{w}) \in \mathcal{R}$, the following is negligible in $\lambda$:

$$\varepsilon_{zk} = \left| \Pr\left[b = 1 \; : \; \begin{array}{l} \mathsf{pp} \leftarrow \mathsf{Gen}(1^\lambda) \\ (\mathsf{x}, \mathsf{w}, \mathsf{st}) \leftarrow \mathcal{A}_1(\mathsf{pp}) \\ \pi \leftarrow \mathsf{Prove}(\mathsf{pp}, \mathsf{x}, \mathsf{w}) \\ b \leftarrow \mathcal{A}_2(\mathsf{st}, \pi) \end{array}\right] - \Pr\left[b = 1 \; : \; \begin{array}{l} (\mathsf{pp}, \mathsf{td}) \leftarrow \mathsf{Sim}_1(1^\lambda) \\ (\mathsf{x}, \mathsf{w}, \mathsf{st}) \leftarrow \mathcal{A}_1(\mathsf{pp}) \\ \pi \leftarrow \mathsf{Sim}_2(\mathsf{td}, \mathsf{x}) \\ b \leftarrow \mathcal{A}_2(\mathsf{st}, \pi) \end{array}\right] \right|.$$

Note that the simulator $\mathsf{Sim}$ does not have access to the witness $\mathsf{w}$; instead, it has the ability to sample a *trapdoor* $\mathsf{td}$ that allows it to generate a proof $\pi$ for any true statement $\mathsf{x}$.

### A.5 Intuition for Zero Knowledge Proofs

In this part, we provide an informal overview of zero knowledge proofs. Consider two parties – a prover and a verifier. The verifier is a color blind person. The prover possesses two objects that are identical in every way except, potentially, their color. The prover claims that the colors of these objects are *different*, but does not want to let the verifier know the colors. How can a color blind verifier check whether the prover's statement is indeed true, i.e., the two otherwise identical objects differ in color? Consider the following protocol:

- First, the prover places both items next to each other on a table.
- Then, the prover leaves the room. The verifier can now flip a coin to decide whether to swap the two objects or not.
- The prover returns and has to declare whether the objects were swapped or not.

At this point, there are two possible scenarios: Either the prover's claim is true and the objects have different colors, or the claim is false and the objects are identical. In the first case, the prover can memorize the color of the left object when leaving the room, and once they return the prover simply checks whether the color of the left object is still the same as before or not. This way, the prover can always pass the check. If, however, the prover was lying and the two objects are completely identical, they have only a $\frac{1}{2}$ chance of *guessing* whether the verifier swapped the two objects or not.

Thus, the protocol is *correct*, in the sense that an honest prover can always pass the check. It is further *sound* (with probability $\frac{1}{2}$), in the sense that an honest verifier has a $50\%$ chance of catching a cheating prover. Finally, it is *zero-knowledge* – it allows the verifier to check the prover's claim without learning anything about the actual colors of the objects. Note that the soundness guarantee can easily be improved by simply repeating the protocol – after $n$ repetitions, the probability that the verifier fails to catch the cheating prover is reduced to only $\frac{1}{2^n}$. Of course, this is only a simplified, illustrative example. Modern zero-knowledge proof systems are complex mathematical protocols with clearly specified assumptions and protocol descriptions, and formal security guarantees. This example nevertheless gives a good idea how a typical zero-knowledge proof can work: The verifier asks the prover to perform a check which does not reveal any additional information except for, potentially, "the prover's statement is incorrect". This check can be repeated multiple times to increase the probability of catching a cheating prover.

Finally, while the example above is an interactive protocol, the majority of modern zero-knowledge protocols are in fact *non-interactive* and *publicly verifiable*, that is, the prover generates a one-shot proof string $\pi$ which can be later checked by *any* verifier without further interacting with the prover. Such protocols require some form of one-time setup phase Blum et al. (1988), which essentially outputs public parameters available for prover and verifier to carry out the protocol. In practice, a setup phase can be conducted by a trusted third party (e.g., government institute) or by some distributed, multi-party computation protocol in order to remove reliance on the trusted third party Kohlweiss et al. (2021).

---

Algorithm 1: Decision Tree Inference

---

**Input:** Decision tree $\mathcal{T}$, input $\mathbf{a}$.
**Output**: Classification result.

1: cur $\leftarrow \mathcal{T}$.root
2: **while** cur is not a leaf **do**
3:     **if** $\mathbf{a}$[cur.attr] $<$ cur.thr **then**
4:         cur $\leftarrow$ cur.left
5:     **else**
6:         cur $\leftarrow$ cur.right
7:     **end if**
8: **end while**
9: **return** cur.class

---

## B    DECISION TREES BACKGROUND

Decision trees are among the most popular machine learning algorithms, particularly known for their effectiveness in classification problems. Their strong performance, combined with high levels of explainability, makes them a popular choice in practice, specifically for fraud detection and automated trading. For simplicity, in the following we focus on binary decision trees for classification problems. The training of a decision tree typically involves recursively splitting the dataset into subsets from the root to the leaves. Each split is determined by a splitting rule that aims to maximize an objective function, such as information gain. The prediction is done by traversing the path from the tree root to the leave, while checking the threshold of the intermediate node and following the corresponding path based on the decision at each step (see Algorithm 1).

## C    RELATED WORK - CONTINUED

Numerous works use cryptographic techniques to certify fairness in machine learning algorithms. For example, a recent work Yadav et al. (2024) focused on privacy-preserving proof of fairness for neural networks. In contrast to our work, Yadav et al. (2024) considers individual fairness, rather than group fairness. Waiwitlikhit et al. (2024) and Zhang et al. (2025) propose a zk system for checking fairness-related bounds for DNNs. Instead of verifying fairness on specific data, Zhang et al. (2025) focus solely on the properties of the models.

Another line of work focuses on proofs for explainability Yadav et al. (2025) and correct training on private data Abbaszadeh et al. (2024); Garg et al. (2023); Sun et al. (2025); Pappas & Papadopoulos (2024).

A recent work by Bourrée et al. (2025) utilizes differentially-private techniques for fairness audits. This work is focused on protecting privacy of the audit dataset, rather than the model.

## D    LIMITATION OF C-PROFITT AGAINST DISHONEST PROVERS

We briefly recap the design of C-PROFITT Shamsabadi et al. (2023). The statement proven by C-PROFITT is roughly: "With respect to some secret test dataset *chosen by the prover*, the model satisfies certain fairness guarantees". This dataset is assumed to be the training data and is thus kept private. If the prover is honest and consistently uses the same data for training and the zero knowledge proof of fairness, then C-PROFITT provides a strong security guarantee. However, it is technically possible for a dishonest prover to use a maliciously crafted test dataset (without anyone noticing). After doing so the prover can prove that *any* (however unfair) model satisfies the given fairness constraints.

While "ZK Proof of Training (ZKPoT)" exists in the literature (i.e. given as a public instance commitments $c_M$ to the trained model and $c_X$ to the training dataset, the prover proves the knowledge of private witness, $M$, $X$ and randomness $r_M, r_X$, such that $c_M = \mathsf{Commit}(M; r_M)$, $c_X = \mathsf{Commit}(X; r_X)$, and $M = \mathsf{Train}(X)$), we show that C-PROFITT does not qualify as ZKPoT

due to the following attack allowing a dishonest prover to convince verifier using an arbitrary DT model.

**Dishonest Prover's Strategy** We describe concrete steps of cheating prover against Algorithm 2 for "ZK proof of demographic parity fair tree training" in C-PROFITT:

1. As inputs, dishonest prover picks an arbitrary decision tree DT and a skewed dataset $X$ of which all the data points have the same sensitive attribute value, i.e., for all $x \in X$, $x[a] = 0$.

2. Since a committed sensitive attribute value $[s]$ is always 0, the counter $c_1$ always remains 0 in Line 9.

3. If $c_1$ is a 0-vector, the fairness metric verification check in Line 11 always passes as the numerator is always 0.

Then the verifier always gets convinced by the above prover, as there is no mechanism to check that the input DT has been actually obtained by executing the correct training algorithm on the input dataset $X$. Analogously, the same strategy works against Algorithm 5 for ZKP for equalized odds-aware tree training, because the counters $c_1$ and $c_3$ would remain 0 which make the conditions in Line 16-17 trivially true.

Thus, ZKP of C-PROFITT clearly does not guarantee that the input DT equals the output of fairness-aware training presented in their Algorithm 1. We observe that the attack stems from the two flaws in the design of C-PROFITT: (1) the ideal functionality $\mathcal{F}_{ZKDT}$ (Figure 6) realized by Algorithm 2 and 5 allows an arbitrary DT to pass the check, because it lets Prover pick a dataset $X$ privately, while the well-formedness of $X$ is never checked, and (2) Algorithm 2 and 5 merely prove that the information gain at each node is below a threshold when DT is evaluated on private $X$ chosen by a dishonest prover, while it is not proving in ZK that each committed split is the best among all possible ones.

# E DETAILS OF INSTANTIATED CRYPTOGRAPHIC PRIMITIVES

## E.1 ADT: OPTIMIZED COMMITMENT FOR DECISION TREES

In Fig.3, we revisit the authenticated decision tree (ADT) structure proposed by Zhang et al. (2020), a variant of the Merkle hash tree designed specifically for decision trees.

To commit to a decision tree $\mathcal{T}$ with $N$ nodes, one first computes the hashes of the identities and class labels of the leaf nodes. Following Group (2024), we instantiate $\bar{\mathsf{H}}$ with the SWIFFT hash function Lyubashevsky et al. (2008). For each non-leaf node $i$, its hash is computed using the hashes of the left (lc) and right (rc) children, along with its identity $v_i$, threshold $v_i.\mathsf{thr}$, attribute $v_i.\mathsf{attr}$, and the identities of its left ($v_i.\mathsf{left}$) and right ($v_i.\mathsf{right}$) children.

The final commitment is obtained by hashing the root hash together with a commitment randomness $\rho$. This extra step is crucial to be able to later prove the zero-knowledge property of our scheme.

To validate a prediction for a single data point evaluated on $\mathcal{T}$, the tree owner provides the prediction path from the root to the corresponding leaf node. The proof also includes the hash of each sibling of every node on the path. Given this information, the verifier simply computes the root hash and compares it against the commitment. This approach optimizes prover time, as proving a prediction's validity requires computing only $O(h)$ hashes, rather than $O(N)$.

Completeness and hiding properties of the ADT commitment scheme are proved in (Zhang et al., 2020, Theorem 3.2). Additionally, we prove the binding property.

**Lemma 1.** *Assuming a hash function instantiating* ADT *is collision resistant,* ADT *is computationally binding.*

*Proof.* Suppose towards a contradiction there exists an adversary finding two openings $\mathcal{T}$ and $\mathcal{T}'$ for the same ADT commitment com. For two distinct decision tree models, it must be that $\mathcal{T}$ and $\mathcal{T}'$ have distinct nodes $v_i \neq v_i'$ for some $i \in [N]$. Denote the hashes of $i$th nodes by $h_i = \bar{\mathsf{H}}(\ldots, v_i, \ldots)$ and $h_i' = \bar{\mathsf{H}}(\ldots, v_i', \ldots)$. If $h_i = h_i'$, then one shows collision resistance of the hash function $\bar{\mathsf{H}}$.

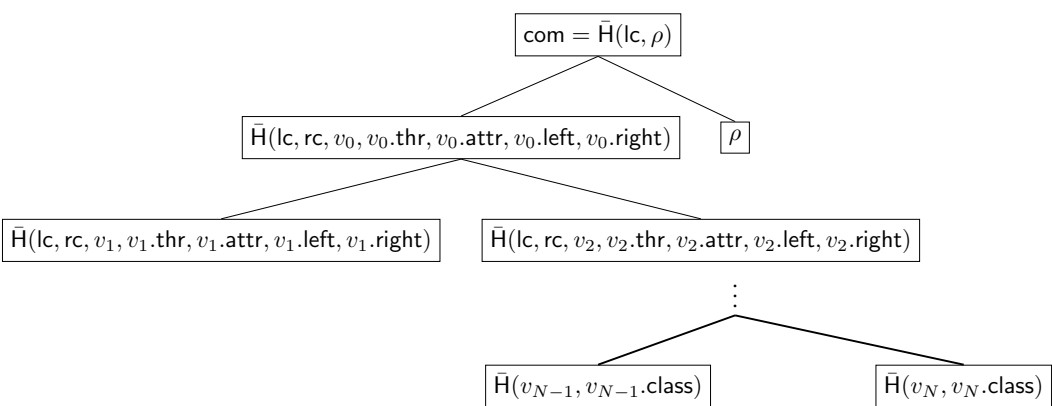

Figure 3: Authenticated decision tree (ADT) commitment Zhang et al. (2020).

If $h_i \neq h_i'$, then it must be that the upper level hashes are distinct for the same reason. Applying this argument iteratively, we have that the root hashes are distinct, contradicting the fact that the adversary has obtained two openings for the same commitment com as the root hash. □

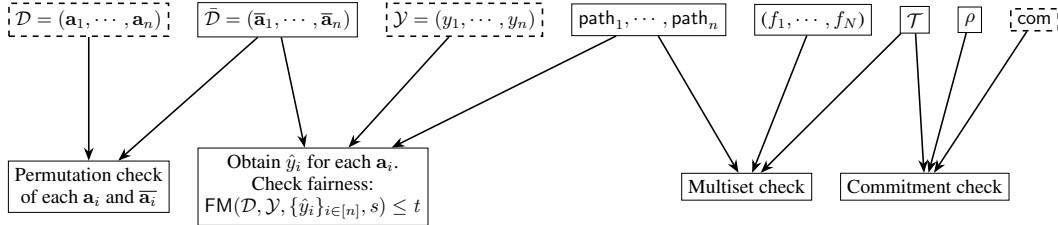

Figure 4: Overview of Zero-Knowledge Decision Tree Fairness adapted from ZKDT Accuracy Zhang et al. (2020). Dashed boxes represent publc inputs to the verifier $\mathcal{V}$ and solid boxes represent private internal values held by the prover $\mathcal{P}^*$.

### E.2 GROTH16 ZERO KNOWLEDGE PROOF FOR DECISION TREE FAIRNESS

Groth16 Groth (2016) is one of the most popular non-interactive zero-knowledge proof systems based on blinear pairings Galbraith et al. (2006), which has perfect completeness, computational and adaptive knowledge soundness in the generic group model Groth (2016) and in the algebraic group model Fuchsbauer et al. (2018), and perfect zero knowledge. It can generate a succinct proof in the form of three group elements, and the verification requires only four pairing operations. Groth16 supports any NP relation in the form of the quadratic arithmetic program (QAP) Gennaro et al. (2013), which is essentially a quadratic polynomial equation defined over a finite field $\mathbb{F}_p$ with exponentially large prime $p$. In this work, we use the *R1CS (rank-1 constraint system)* representation of QAPs. Let $\mathbf{A}, \mathbf{B}, \mathbf{C} \in \mathbb{F}_p^{n \times m}$ be public matrices, where each row specifies quadratic constraints over $m$ variables. Given such matrices, a valid instance of R1CS is a pair $(\mathsf{x}, \mathsf{w})$ where $\mathsf{x} \in \mathbb{F}_p^l$ is a public input and $\mathsf{w} \in \mathbb{F}_p^{m-l}$ is a private witness such that for $\mathbf{z} = (\mathsf{x}, \mathsf{w})$,

$$\mathbf{Az} \circ \mathbf{Bz} = \mathbf{Cz} \bmod p \tag{2}$$

Thus, proving the $\mathcal{R}_{\mathsf{cp}}$ relation boils down to defining suitable R1CS matrices and vectors. In Zhang et al. (2020) and the accompanying implementation Group (2024), the authors provided an efficient approach to proving (1) opening $\mathcal{T}$ of ADT commitment ("Commitment check" in Fig. 4), and (2) batched decision tree prediction $\hat{y}_i = \mathcal{T}(\mathbf{a}_i)$ for $i = 1, \ldots, n$ ("Permutation check" and 'Multiset check'" in Fig. 4). Unlike Zhang et al. (2020), we additionally prove $\mathsf{FM}(\mathcal{D}, \mathcal{Y}, \hat{\mathcal{Y}}, s) \leq t$ in

the form of an R1CS relation. In the case of Demographic Parity, recall that the fairness metric $\mathsf{FM}(\mathcal{D}, \mathcal{Y}, \hat{\mathcal{Y}}, s)$ is defined as:

$$\left| \frac{\sum_{i \in I_0} \hat{y}_i}{n_0} - \frac{\sum_{i \in I_1} \hat{y}_i}{n_1} \right| \leq t$$

where $I_0 = \{i \in [n] : \mathbf{a}_i[s] = 0\}$, $I_1 = \{i \in [n] : \mathbf{a}_i[s] = 1\}$, $n_0 = |I_0|$, $n_1 = |I_1|$, and $\hat{y}_i \in \{0, 1\}$ are available as a subvector of $\mathbf{z}$ as a result of Step (2) above. We transform it into the following equivalent condition:

$$w_0 = \sum_{i \in I_0} c \cdot n_1 \cdot \hat{y}_i \quad w_1 = \sum_{i \in I_1} c \cdot n_0 \cdot \hat{y}_i$$

$$|w_0 - w_1| \leq c \cdot t \cdot n_0 \cdot n_1$$

where $c$ is a a suitable scaling constant such that $c \cdot t \in \mathbb{Z}$ (which we need to express the constraint as a equation in $\mathbb{F}_p$ since $t$ is typically a small rational number). The first two conditions are quadratic equations which can be expressed as R1CS constraints (Note also that $n_0, n_1, I_0, I_1$ are public since they can be derived from $(\mathcal{D}, \mathcal{Y})$). The third equation can be transformed into R1CS by invoking the comparison gadget available in Group (2024).

# F  SECURITY OF OUR CONSTRUCTION

In this part, we provide formal definitions and proof of security of our generic construction $\Pi$.

## F.1  FORMAL DEFINITION OF SECURITY PROPERTIES

We define the following security properties for our construction $\Pi$:

**Completeness** requires that if both the Prover and Verifier honestly follow the protocol, and the Prover owns a fair model $M$, then the Verifier is always convinced *and* they agree on the same synthetic data.

Formally, let $M$ be a model satisfying the following condition with overwhelming probability: for $r \xleftarrow{\$} \{0,1\}^k$, $(\mathcal{D}, \mathcal{Y}) \leftarrow \mathsf{DataGen}(r)$, and $\hat{y} = M(\mathcal{D})$, we have $\mathsf{FM}(\mathcal{D}, \mathcal{Y}, \hat{y}, s) \leq t$. $\Pi$ is *complete* if the following game outputs 1 with overwhelming probability:

1: $\mathsf{pp} \leftarrow \mathsf{Setup}(1^\lambda)$
2: $\mathsf{com} \xleftarrow{\rho} \mathsf{Commit}(M)$
3: $((\mathcal{D}_\mathcal{P}, \mathcal{Y}_\mathcal{P}), (\mathcal{D}_\mathcal{V}, \mathcal{Y}_\mathcal{V}, b)) \leftarrow \langle \mathcal{P}(M, \rho), \mathcal{V} \rangle(\mathsf{pp}, \mathsf{com})$
4: **Output** $(b = 1) \wedge (\mathcal{D}_\mathcal{P} = \mathcal{D}_\mathcal{V}) \wedge (\mathcal{Y}_\mathcal{P} = \mathcal{Y}_\mathcal{V})$

**Binding** requires that a commitment cannot be opened to different models. Formally, $\Pi$ is *binding* against dishonest model providers if for any PPT adversary $\mathcal{A}$, the following game outputs 1 with probability negligible in $\lambda$:

1: $(\mathsf{com}, M, \rho, M', \rho') \leftarrow \mathcal{A}(1^\lambda)$
2: **Output** $(M \neq M') \wedge (\mathsf{Open}(\mathsf{com}, M, \rho) = \mathsf{Open}(\mathsf{com}, M', \rho') = 1)$

**Dishonest Provider Robustness** requires that if a potentially dishonest Prover convinces an honest Verifier, then the Prover must *know* a model that satisfies the fairness criteria with respect to the test dataset generated during the interactive protocol. Formally, $\Pi$ has *dishonest provider robustness* against dishonest model providers if for any PPT adversary $\mathcal{P}^* = (\mathcal{P}_0^*, \mathcal{P}_1^*)$, there exists a (non-blackbox) polynomial time extractor $\mathcal{E}_{\mathcal{P}^*}$ such that the following game outputs 1 with probability negligible in $\lambda$:

1: $\mathsf{pp} \leftarrow \mathsf{Setup}(1^\lambda)$
2: $(\mathsf{com}, \mathsf{st}) \leftarrow \mathcal{P}_0^*(\mathsf{pp})$
3: $(*, (\mathcal{D}_\mathcal{V}, \mathcal{Y}_\mathcal{V}, b)) \leftarrow \langle \mathcal{P}_1^*(\mathsf{st}), \mathcal{V} \rangle(\mathsf{pp}, \mathsf{com})$
4: $(M, \rho) \leftarrow \mathcal{E}_{\mathcal{P}^*}(\mathsf{pp})$
5: **return** $(b = 1) \wedge (((\mathcal{D}_\mathcal{V}, \mathcal{Y}_\mathcal{V}), M) \notin \mathcal{R}_{\mathsf{fair}} \vee \mathsf{Open}(\mathsf{com}, M, \rho) \neq 1)$

**Model Privacy** requires that a potentially dishonest Verifier learns nothing about the Prover's model, except that it satisfies the relation $\mathcal{R}_{\mathsf{fair}}$. Formally, let $M$ be a model satisfying the following condition with overwhelming probability: for $r \xleftarrow{\$} \{0,1\}^k$, $(\mathcal{D}, \mathcal{Y}) \leftarrow \mathsf{DataGen}(r)$, and $\hat{y} = M(\mathcal{D})$, we have $\mathsf{FM}(\mathcal{D}, \mathcal{Y}, \hat{y}, s) \leq t$. For an arbitrary interactive PPT algorithm $\mathcal{V}^*$, let $\mathsf{view}_{\mathcal{V}^*}^{\mathcal{P}}(M)$ be the distribution of the following information: $\mathsf{pp} \leftarrow \mathsf{Setup}(1^\lambda)$, $\mathsf{com} \xleftarrow{\rho} \mathsf{Commit}(M)$ and all the incoming messages that $\mathcal{V}^*$ receives from $\mathcal{P}$ during an execution offered $\langle \mathcal{P}(M, \rho), \mathcal{V}^* \rangle(\mathsf{pp}, \mathsf{com})$.

$\Pi$ has *model privacy* against dishonest verifiers if for any PPT $\mathcal{V}^*$, there exists a PPT simulator $\mathcal{S}$ such that the output of $\mathcal{S}$ is indistinguishable from the distribution $\mathsf{view}_{\mathcal{V}^*}^{\mathcal{P}}(M)$.

**Test Data Transparency** requires that neither party can bias the distribution of the test dataset generated during the interactive protocol, provided the protocol does not abort. Formally, let $\mathsf{data}_{\mathcal{V}}^{\mathcal{P}^*}$ (resp. $\mathsf{data}_{\mathcal{P}}^{\mathcal{V}^*}$) be the distribution of $(\mathcal{D}_{\mathcal{V}}, \mathcal{Y}_{\mathcal{V}})$ (resp. $(\mathcal{D}_{\mathcal{P}}, \mathcal{Y}_{\mathcal{P}})$) generated by $\mathcal{V}$ (resp. $\mathcal{P}$) after interacting with a potentially malicious prover $\mathcal{P}^*$ (resp. verifier $\mathcal{V}^*$), conditioned on the protocol not aborting. $\Pi$ has *test data transparency* if the following conditions hold:

1. for any PPT cheating $\mathcal{P}^*$, $\mathsf{data}_{\mathcal{V}}^{\mathcal{P}^*}$ is indistinguishable with $(\mathcal{D}, \mathcal{Y}) \leftarrow \mathsf{DataGen}(r)$ for $r \xleftarrow{\$} \{0,1\}^k$.

2. for any PPT cheating $\mathcal{V}^*$, $\mathsf{data}_{\mathcal{P}}^{\mathcal{V}^*}$ is indistinguishable with $(\mathcal{D}, \mathcal{Y}) \leftarrow \mathsf{DataGen}(r)$ for $r \xleftarrow{\$} \{0,1\}^k$.

## F.2 Proof of Security

We state a formal version of the security theorem of our construction $\Pi$.

**Theorem 1.** *We have the following security properties of Protocol 1:*

- *If ZK and COM are complete, Protocol 1 is complete.*

- *If COM is binding, Protocol 1 is binding against dishonest model owners.*

- *If ZK is knowledge sound, Protocol 1 has dishonest provider robustness.*

- *If ZK is zero knowledge, COM is hiding, H is modeled as a programmable random oracle, and an adversary makes at most $poly(\lambda)$ queries to H, Protocol 1 has model privacy against dishonest verifiers.*

- *If the same conditions as model privacy hold, Protocol 1 satisfies test data transparency.*

*Proof.* **Completeness** follows by inspection. If $\mathcal{P}$ and $\mathcal{V}$ are honest, then they derive the same $(\mathcal{D}, \mathcal{Y}) \leftarrow \mathsf{DataGen}(r)$ with $r = r_1 \oplus r_2$. Moreover, since $r$ is uniformly random in $\{0,1\}^k$, $((\mathcal{D}, \mathcal{Y}), M) \in \mathcal{R}_{\mathsf{fair}}$ with overwhelming probability. Finally, since COM and ZK are complete, $\mathcal{P}$ outputs a valid commitment $\mathsf{com} = \mathsf{ADT.Commit}(M, \rho)$ and a valid proof $\pi = \mathsf{ZK.Prove}(\mathsf{pp}, (\mathsf{com}, \mathcal{D}, \mathcal{Y}), (M, \rho))$ such that $\mathsf{ZK.V}(\mathsf{pp}, (\mathsf{com}, \mathcal{D}, \mathcal{Y}), \pi) = 1$

**Binding** immediately follows from the binding property of COM.

**Dishonest Provider Robustness** Given an adversary $\mathcal{A}$ against the dishonest provider robustness game, we first construct a cheating prover $\mathcal{P}'$ against the adaptive knowledge soundness game. Consider the following $\mathcal{P}'$:

1. Upon receiving $\mathsf{pp}$ as input, forward $\mathsf{pp}$ to $\mathcal{P}_0^*$

2. Upon receiving $\mathsf{com}$ from $\mathcal{P}_0^*$, conduct an interactive process $(*, (\mathcal{D}_{\mathcal{V}}, \mathcal{Y}_{\mathcal{V}}, b)) \leftarrow \langle \mathcal{P}_1^*(\mathsf{st}), \mathcal{V} \rangle(\mathsf{pp}, \mathsf{com})$ by emulating an honest verifier $\mathcal{V}$.

3. Upon receiving $\pi$ from $\mathcal{P}_1^*$, set $\mathsf{x} = (\mathsf{com}, \mathcal{D}_{\mathcal{V}}, \mathcal{Y}_{\mathcal{V}})$ and output $(\mathsf{x}, \pi)$.

Clearly, if $\mathcal{V}$ halts by outputting $b = 1$, then $\mathsf{ZK.Verify}$ accepts $(\mathsf{x}, \pi)$. As such, there exists a knowledge extractor $\mathcal{E}_{\mathcal{P}'}$ for ZK, which obtains a valid witness $(M, \rho)$ such that $(\mathsf{x}, (M, \rho)) \in \mathcal{R}_{\mathsf{cp}}$ by

running internally $\mathcal{P}'$ causing ZK.$\mathcal{V}$ to accept. Clearly, $(x, (M, \rho)) \in \mathcal{R}_{cp}$ implies $((\mathcal{D}_\mathcal{V}, \mathcal{Y}_\mathcal{V}), M) \in \mathcal{R}_{fair}$ and $\mathsf{Open}(\mathsf{com}, M, \rho) = 1$, meaning that $\mathcal{E}_{\mathcal{P}'}$ in combination with the procedures of $\mathcal{P}'$ serves as a valid extractor $\mathcal{E}_{\mathcal{P}^*}$ in the dishonest provider robustness game.

**Model Privacy** We prove that the view of $\mathsf{view}_{\mathcal{V}^*}^{\mathcal{P}}(M)$ can be simulated by a simulator $\mathcal{S}$ that does not know $M$. The detailed hybrids are presented in Algorithm 2. Let $\mathcal{A}$ be an distinguisher. Below we denote by $p_i(\lambda)$ the probability that $\mathcal{A}$ outputs 1 given the view of the verifier $\mathcal{V}^*$ in Hybrid-$i$. We first consider Hybrid-0, a real execution of the protocol where $\mathcal{P}$ is honest and $\mathcal{V}^*$ is a potentially malicious verifier.

In Hybrid-1, we replace the zero knowledge proof $\pi$ and public parameters pp with simulated ones. Let ZK.$(\mathsf{Sim}_1, \mathsf{Sim}_2)$ be a zero knowledge simulator for ZK. Since ZK is zero knowledge, the simulated view of $\mathcal{V}^*$ in Hybrid-1 is indistinguishable from that of Hybrid-0, that is,

$$|p_1(\lambda) - p_0(\lambda)| \leq \varepsilon_{zk}$$

In Hybrid-2, we replace $h_1$ with a random string $h_1 \xleftarrow{\$} \{0,1\}^\ell$ and program the random oracle H such that $\mathsf{H}(r_1) = h_1$. Since the adversary makes at most $poly(\lambda)$ queries to H, programming here fails with probability at most $poly(\lambda)/2^{-k}$. Thus, we have that

$$|p_2(\lambda) - p_1(\lambda)| \leq poly(\lambda)/2^{-k}.$$

Assuming $k \in \Omega(\lambda)$, Hybrid-2 and Hybrid-1 are statistically indistinguishable.

In Hybrid-3 we sample $r \xleftarrow{\$} \{0,1\}^k$ first and set $r_1 := r \oplus r_2$ using received $r_2$. Since the distribution of $r$ in Hybrid-2 is already determined to be uniform at Step 5, Hybrid-3 and Hybrid-2 are perfectly indistinguishable, that is,

$$p_3(\lambda) = p_2(\lambda)$$

In Hybrid-4, we skip the check of the fairness condition. Since $r$ is uniformly random at this stage and a real model $M$ passes the fairness check with overwhelming probability, skipping the fairness check in Hybrid-4 does not affect the view of $\mathcal{V}^*$. Thus, we have that Hybrid-4 and Hybrid-3 are statistically indistinguishable:

$$|p_4(\lambda) - p_3(\lambda)| \leq negl(\lambda)$$

In Hybrid-5, we replace the simulated commitment to real $M$ with a simulated commitment to a dummy string 0. By the hiding property of the commitment scheme, the simulated commitment is indistinguishable from a real one. Thus, Hybrid-5 and Hybrid-4 are indistinguishable.

$$|p_5(\lambda) - p_4(\lambda)| \leq \varepsilon_{hiding}$$

Since Hybrid-5 does not depend on the model $M$, we can conclude that the procedure of Hybrid-5 can be used as a simulator $\mathcal{S}$ for $\mathsf{view}_{\mathcal{V}^*}^{\mathcal{P}}(M)$.

**Test Data Transparency** We first note that the dishonest verifier case (i.e., $\mathsf{data}_{\mathcal{P}}^{\mathcal{V}^*}$ is indistinguishable with $(\mathcal{D}, \mathcal{Y}) \leftarrow \mathsf{DataGen}(r)$ for $r \xleftarrow{\$} \{0,1\}^k$) is already implied by the proof of model privacy. That is, in Hybrid-5, $\mathcal{P}$ locally outputs the test data $\mathcal{D}_\mathcal{P}$ and labels $\mathcal{Y}_\mathcal{P}$ obtained by running $\mathsf{DataGen}(r)$ on uniformly random $r$.

The other case (i.e., $\mathsf{data}_\mathcal{V}^{\mathcal{P}^*}$ is indistinguishable with $(\mathcal{D}, \mathcal{Y}) \leftarrow \mathsf{DataGen}(r)$ for $r \xleftarrow{\$} \{0,1\}^k$) can be proved analogously without relying on ZK and hiding. For completeness, we present the hybrids in Algorithm 3.

**Algorithm 2: Hybrids for Model Privacy of Theorem 1**

$\mathsf{Hyb}_0(\lambda)$

1: $\mathsf{pp} \leftarrow \mathsf{Setup}(1^\lambda)$ and $\mathsf{com} \xleftarrow{\rho} \mathsf{Commit}(M)$
2: $\mathcal{V}^*(\mathsf{pp}, \mathsf{com})$ sends $h_2$
3: $\mathcal{P}$ sends $h_1 = \mathsf{H}(r_1)$, where $r_1 \leftarrow \{0,1\}^k$
4: $\mathcal{V}^*$ sends $r_2$
5: $\mathcal{P}$ checks that $h_2 = \mathsf{H}(r_2)$ and does:

    (a) $r \leftarrow r_1 \oplus r_2$
    (b) $(\mathcal{D}_\mathcal{P}, \mathcal{Y}_\mathcal{P}) \leftarrow \mathsf{DataGen}(r)$
    (c) check $\mathsf{FM}(\mathcal{D}_\mathcal{P}, \mathcal{Y}_\mathcal{P}, M(\mathcal{D}_\mathcal{P}), s) \le t$
    (d) $\pi \leftarrow \mathsf{ZK.Prove}(\mathsf{pp}, (\mathsf{com}, \mathcal{D}_\mathcal{P}, \mathcal{Y}_\mathcal{P}), (M, \rho))$
    (e) send $(\pi, r_1)$ to $\mathcal{V}^*$
    (f) locally output $(\mathcal{D}_\mathcal{P}, \mathcal{Y}_\mathcal{P})$

Hybrid-1

1: $(\mathsf{pp}, \mathsf{td}) \leftarrow \mathsf{ZK.Sim}_1(1^\lambda)$ and $\mathsf{com} \xleftarrow{\rho} \mathsf{Commit}(M)$
2: $\mathcal{V}^*(\mathsf{pp}, \mathsf{com})$ sends $h_2$
3: $\mathcal{P}$ sends $h_1 = \mathsf{H}(r_1)$, where $r_1 \leftarrow \{0,1\}^k$
4: $\mathcal{V}^*$ sends $r_2$
5: $\mathcal{P}$ checks that $h_2 = \mathsf{H}(r_2)$ and does:

    (a) $r \leftarrow r_1 \oplus r_2$
    (b) $(\mathcal{D}_\mathcal{P}, \mathcal{Y}_\mathcal{P}) \leftarrow \mathsf{DataGen}(r)$
    (c) check $\mathsf{FM}(\mathcal{D}_\mathcal{P}, \mathcal{Y}_\mathcal{P}, M(\mathcal{D}_\mathcal{P}), s) \le t$
    (d) $\pi \leftarrow \mathsf{ZK.Sim}_2(\mathsf{td}, (\mathsf{com}, \mathcal{D}_\mathcal{P}, \mathcal{Y}_\mathcal{P}))$
    (e) send $(\pi, r_1)$ to $\mathcal{V}^*$
    (f) locally output $(\mathcal{D}_\mathcal{P}, \mathcal{Y}_\mathcal{P})$

Hybrid-2

1: $(\mathsf{pp}, \mathsf{td}) \leftarrow \mathsf{ZK.Sim}_1(1^\lambda)$ and $\mathsf{com} \xleftarrow{\rho} \mathsf{Commit}(M)$
2: $\mathcal{V}^*(\mathsf{pp}, \mathsf{com})$ sends $h_2$
3: $\mathcal{P}$ sends $h_1 \leftarrow \{0,1\}^\ell$
4: $\mathcal{V}^*$ sends $r_2$
5: $\mathcal{P}$ checks that $h_2 = \mathsf{H}(r_2)$. $\mathcal{P}$ samples $r_1 \xleftarrow{\$} \{0,1\}^k$ and programs the random oracle such that $h_1 = \mathsf{H}(r_1)$ (or aborts if it fails to program). $\mathcal{P}$ does:

    (a) $r \leftarrow r_1 \oplus r_2$
    (b) $(\mathcal{D}_\mathcal{P}, \mathcal{Y}_\mathcal{P}) \leftarrow \mathsf{DataGen}(r)$
    (c) check $\mathsf{FM}(\mathcal{D}_\mathcal{P}, \mathcal{Y}_\mathcal{P}, M(\mathcal{D}_\mathcal{P}), s) \le t$
    (d) $\pi \leftarrow \mathsf{ZK.Sim}_2(\mathsf{td}, (\mathsf{com}, \mathcal{D}_\mathcal{P}, \mathcal{Y}_\mathcal{P}))$
    (e) send $(\pi, r_1)$ to $\mathcal{V}^*$
    (f) locally output $(\mathcal{D}_\mathcal{P}, \mathcal{Y}_\mathcal{P})$

Hybrid-3

1: $(\mathsf{pp}, \mathsf{td}) \leftarrow \mathsf{ZK.Sim}_1(1^\lambda)$ and $\mathsf{com} \xleftarrow{\rho} \mathsf{Commit}(M)$
2: $\mathcal{V}^*(\mathsf{pp}, \mathsf{com})$ sends $h_2$
3: $\mathcal{P}$ sends $h_1 \leftarrow \{0,1\}^\ell$
4: $\mathcal{V}^*$ sends $r_2$
5: $\mathcal{P}$ checks that $h_2 = \mathsf{H}(r_2)$. $\mathcal{P}$ samples $r \xleftarrow{\$} \{0,1\}^k$, sets $r_1 := r \oplus r_2$ and programs the random oracle such that $h_1 = \mathsf{H}(r_1)$ (or aborts if it fails to program). $\mathcal{P}$ does:

    (a) $\cancel{r \leftarrow r_1 \oplus r_2}$
    (b) $(\mathcal{D}_\mathcal{P}, \mathcal{Y}_\mathcal{P}) \leftarrow \mathsf{DataGen}(r)$
    (c) check $\mathsf{FM}(\mathcal{D}_\mathcal{P}, \mathcal{Y}_\mathcal{P}, M(\mathcal{D}_\mathcal{P}), s) \le t$
    (d) $\pi \leftarrow \mathsf{ZK.Sim}_2(\mathsf{td}, (\mathsf{com}, \mathcal{D}_\mathcal{P}, \mathcal{Y}_\mathcal{P}))$
    (e) send $(\pi, r_1)$ to $\mathcal{V}^*$
    (f) locally output $(\mathcal{D}_\mathcal{P}, \mathcal{Y}_\mathcal{P})$

Hybrid-4

1: $(\mathsf{pp}, \mathsf{td}) \leftarrow \mathsf{ZK.Sim}_1(1^\lambda)$ and $\mathsf{com} \xleftarrow{\rho} \mathsf{Commit}(M)$
2: $\mathcal{V}^*(\mathsf{pp}, \mathsf{com})$ sends $h_2$
3: $\mathcal{P}$ sends $h_1 \leftarrow \{0,1\}^\ell$
4: $\mathcal{V}^*$ sends $r_2$
5: $\mathcal{P}$ checks that $h_2 = \mathsf{H}(r_2)$. $\mathcal{P}$ samples $r \xleftarrow{\$} \{0,1\}^k$, sets $r_1 := r \oplus r_2$ and programs the random oracle such that $h_1 = \mathsf{H}(r_1)$ (or aborts if it fails to program). $\mathcal{P}$ does:

    (a) $\cancel{r \leftarrow r_1 \oplus r_2}$
    (b) $(\mathcal{D}_\mathcal{P}, \mathcal{Y}_\mathcal{P}) \leftarrow \mathsf{DataGen}(r)$

    (c) check $\cancel{\mathsf{FM}(\mathcal{D}_\mathcal{P}, \mathcal{Y}_\mathcal{P}, M(\mathcal{D}_\mathcal{P}), s) \le t}$
    (d) $\pi \leftarrow \mathsf{ZK.Sim}_2(\mathsf{td}, (\mathsf{com}, \mathcal{D}_\mathcal{P}, \mathcal{Y}_\mathcal{P}))$
    (e) send $(\pi, r_1)$ to $\mathcal{V}^*$
    (f) locally output $(\mathcal{D}_\mathcal{P}, \mathcal{Y}_\mathcal{P})$

Hybrid-5

1: $(\mathsf{pp}, \mathsf{td}) \leftarrow \mathsf{ZK.Sim}_1(1^\lambda)$ and $\mathsf{com} \xleftarrow{\rho} \mathsf{Commit}(0)$
2: $\mathcal{V}^*(\mathsf{pp}, \mathsf{com})$ sends $h_2$
3: $\mathcal{P}$ sends $h_1 \leftarrow \{0,1\}^\ell$
4: $\mathcal{V}^*$ sends $r_2$
5: $\mathcal{P}$ checks that $h_2 = \mathsf{H}(r_2)$. $\mathcal{P}$ samples $r \xleftarrow{\$} \{0,1\}^k$, sets $r_1 := r \oplus r_2$ and programs the random oracle such that $h_1 = \mathsf{H}(r_1)$ (or aborts if it fails to program). $\mathcal{P}$ does:

    (a) $\cancel{r \leftarrow r_1 \oplus r_2}$
    (b) $(\mathcal{D}_\mathcal{P}, \mathcal{Y}_\mathcal{P}) \leftarrow \mathsf{DataGen}(r)$

    (c) check $\cancel{\mathsf{FM}(\mathcal{D}_\mathcal{P}, \mathcal{Y}_\mathcal{P}, M(\mathcal{D}_\mathcal{P}), s) \le t}$
    (d) $\cancel{\pi \leftarrow \mathsf{ZK.Sim}_2(\mathsf{td}, (\mathsf{com}, \mathcal{D}_\mathcal{P}, \mathcal{Y}_\mathcal{P}))}$
    (e) send $(\pi, r_1)$ to $\mathcal{V}^*$
    (f) locally output $(\mathcal{D}_\mathcal{P}, \mathcal{Y}_\mathcal{P})$

Transition from Hybrid-0 to Hybrid-1 is similar to the transition from Hybrid-1 to Hybrid-2 in the proof of model privacy.

In Hybrid-2, upon receiving $h_1$ from $\mathcal{P}^*$, $\mathcal{V}$ extracts its preimage $r'_1$ by observing the queries made by $\mathcal{P}^*$ to the random oracle H. $\mathcal{V}$ then aborts if one of the following events happens: 1) it either fails to extract $r'_1$ but $\mathcal{P}^*$ later manages to open $h_1$ to some $r_1$ such that $\mathsf{H}(r_1) = h_1$; 2) it extracts $r'_1$ but $\mathcal{P}^*$ later manages to open $h_1$ to $r'_1 \neq r_1$.

Case 1) implies that one of the random oracle responses hits $h_1$, which happens with probability at most $poly(\lambda)/2^\ell$ due to the uniformity of the random oracle responses and the assumption that $\mathcal{P}^*$ makes at most $poly(\lambda)$ queries.

Case 2) implies that $\mathcal{P}^*$ manages to find a collision in one of the $poly(\lambda)$ random oracle responses, which happens with probability at most $poly(\lambda)/2^\ell$ by the birthday bound. Thus, the probability that $\mathcal{V}$ aborts is at most $poly(\lambda)/2^\ell$.

Overall, Hybrid-2 and Hybrid-1 are statistically indistinguishable.

$$|p_2(\lambda) - p_1(\lambda)| \leq negl(\lambda)$$

Transition from Hybrid-2 to Hybrid-3 is similar to the transition from Hybrid-2 to Hybrid-3 in the proof of model privacy.

Clearly, in in Hybrid-2, $\mathcal{V}$ locally outputs the test data $\mathcal{D}_\mathcal{V}$ and labels $\mathcal{Y}_\mathcal{V}$ obtained by running $\mathsf{DataGen}(r)$ on uniformly random $r$, assuming that $\mathcal{P}^*$ does not abort. $\qquad\square$

**Remark 2.** *Since $\mathcal{P}^*$ opens $h_1$ after receiving $r_2$ from $\mathcal{V}$, it might abort based on the value of $r = r_1 \oplus r_2$ if it does not like the corresponding test data $\mathcal{D}$ and labels $\mathcal{Y}$. Thus, we obtain the so-called* security with abort *(see e.g., Lindell (2009)) for test data transparency against adversarial model owners. In our setting, security with abort is sufficient by having the verifier $\mathcal{V}$ reject the model whenever $\mathcal{P}^*$ suspiciously causes the protocol to abort.*

# G    DETAILS OF EXPERIMENTS AND ADDITIONAL EVALUATIONS

In this section, we present additional experiments that were not included in the main body of the paper. These experiments further validate the performance and effectiveness of our proposed AudiFair protocol for checking fairness in decision trees.

## G.1    SOURCE CODE AND EXPERIMENTAL SETUP

We implemented the COM and ZK components of our AudiFair protocol in C++, utilizing the lib-snark library Lab (2014) for zero-knowledge proofs and forking the code for proof of accuracy of decision trees Zhang et al. (2020). The SHA3-256 hash function and the synthetic data generation algorithm are implemented in Python, using the hashlib and SDV libraries Developers (2024), respectively. Version information is available in environment.yml.

The experiments were conducted on an Amazon EC2 c7a.12xlarge instance with 96GB RAM, which allowed us to parallelize the computation of FFT and elliptic curve operations during the Groth16 NIZK Setup and Prove phases.

## G.2    DATA SETS

For our experiments, we first trained decision trees of height $h = 10$ using scikit-learn Pedregosa et al. (2011) on the following datasets:

- Default Credit ($d = 23$) Repository (2009): The ground truth label indicates whether a person will default on an credit card payment, and 22% of the datapoints belong to the default class. We use the gender (Male, Female) as a sensitive attribute, with 40% of the datapoints being Males.

---

**Algorithm 3: Hybrids for Test Data Transparency of Theorem 1**

Hybrid-0

1: $\mathsf{pp} \leftarrow \mathsf{Setup}(1^\lambda)$
2: $\mathcal{P}^*(\mathsf{pp})$ sends com
3: $\mathcal{V}$ sends $h_2 = \mathsf{H}(r_2)$, where $r_2 \leftarrow \{0,1\}^k$
4: $\mathcal{P}^*$ sends $h_1$
5: $\mathcal{V}$ sends $r_2$
6: $\mathcal{P}^*$ sends $(\pi, r_1)$
7: $\mathcal{V}$ checks that $h_1 = \mathsf{H}(r_1)$ and does:
    (a) $r \leftarrow r_1 \oplus r_2$
    (b) $(\mathcal{D}_\mathcal{V}, \mathcal{Y}_\mathcal{V}) \leftarrow \mathsf{DataGen}(r)$
    (c) run $b \leftarrow \mathsf{ZK.Verify}(\mathsf{pp}, (\mathsf{com}, \mathcal{D}_\mathcal{V}, \mathcal{Y}_\mathcal{V}), \pi)$
    (d) locally output $(\mathcal{D}_\mathcal{V}, \mathcal{Y}_\mathcal{V}, b)$

Hybrid-1

1: $\mathsf{pp} \leftarrow \mathsf{Setup}(1^\lambda)$
2: $\mathcal{P}^*(\mathsf{pp})$ sends com
3: $\mathcal{V}$ sends $h_2 \xleftarrow{\$} \{0,1\}^\ell$
4: $\mathcal{P}^*$ sends $h_1$
5: $\mathcal{V}$ samples $r_2 \xleftarrow{\$} \{0,1\}^k$ and programs the random oracle such that $h_2 = \mathsf{H}(r_2)$ (or aborts if it fails to program). $\mathcal{V}$ sends $r_2$
6: $\mathcal{P}^*$ sends $(\pi, r_1)$
7: $\mathcal{V}$ checks that $h_1 = \mathsf{H}(r_1)$ and does:
    (a) $r \leftarrow r_1 \oplus r_2$
    (b) $(\mathcal{D}_\mathcal{V}, \mathcal{Y}_\mathcal{V}) \leftarrow \mathsf{DataGen}(r)$
    (c) run $b \leftarrow \mathsf{ZK.Verify}(\mathsf{pp}, (\mathsf{com}, \mathcal{D}_\mathcal{V}, \mathcal{Y}_\mathcal{V}), \pi)$
    (d) locally output $(\mathcal{D}_\mathcal{V}, \mathcal{Y}_\mathcal{V}, b)$

Hybrid-2

1: $\mathsf{pp} \leftarrow \mathsf{Setup}(1^\lambda)$
2: $\mathcal{P}^*(\mathsf{pp})$ sends com
3: $\mathcal{V}$ sends $h_2 \xleftarrow{\$} \{0,1\}^\ell$
4: $\mathcal{P}^*$ sends $h_1$
5: $\mathcal{V}$ extracts $r_1'$ such that $h_1 = \mathsf{H}(r_1')$ by observing queries to the random oracle $\mathsf{H}$ if it exists in the query history; else, $\mathcal{V}$ sets $r_1' := \bot$. $\mathcal{V}$ samples $r_2 \xleftarrow{\$} \{0,1\}^k$ and programs the random oracle such that $h_2 = \mathsf{H}(r_2)$ (or aborts if it fails to program). $\mathcal{V}$ sends $r_2$
6: $\mathcal{P}^*$ sends $(\pi, r_1)$
7: $\mathcal{V}$ checks that $h_1 = \mathsf{H}(r_1)$. If the check passes and $r_1' \neq r_1$, $\mathcal{V}$ aborts. $\mathcal{V}$ does:
    (a) $r \leftarrow r_1 \oplus r_2$
    (b) $(\mathcal{D}_\mathcal{V}, \mathcal{Y}_\mathcal{V}) \leftarrow \mathsf{DataGen}(r)$
    (c) run $b \leftarrow \mathsf{ZK.Verify}(\mathsf{pp}, (\mathsf{com}, \mathcal{D}_\mathcal{V}, \mathcal{Y}_\mathcal{V}), \pi)$
    (d) locally output $(\mathcal{D}_\mathcal{V}, \mathcal{Y}_\mathcal{V}, b)$

Hybrid-3

1: $\mathsf{pp} \leftarrow \mathsf{Setup}(1^\lambda)$
2: $\mathcal{P}^*(\mathsf{pp})$ sends com
3: $\mathcal{V}$ sends $h_2 \xleftarrow{\$} \{0,1\}^\ell$
4: $\mathcal{P}^*$ sends $h_1$
5: $\mathcal{V}$ extracts $r_1'$ such that $h_1 = \mathsf{H}(r_1')$ by observing queries to the random oracle $\mathsf{H}$ if it exists in the query history; else, $\mathcal{V}$ sets $r_1' := \bot$. If $r_1' \neq \bot$, $\mathcal{V}$ samples $r \xleftarrow{\$} \{0,1\}^k$, and sets $r_2 := r \oplus r_1'$; else $\mathcal{V}$ samples $r_2 \xleftarrow{\$} \{0,1\}^k$. $\mathcal{V}$ programs the random oracle such that $h_2 = \mathsf{H}(r_2)$ (or aborts if it fails to program); $\mathcal{V}$ sends $r_2$
6: $\mathcal{P}^*$ sends $(\pi, r_1)$
7: $\mathcal{V}$ checks that $h_1 = \mathsf{H}(r_1)$. If the check passes and $r_1' \neq r_1$, $\mathcal{V}$ aborts. $\mathcal{V}$ does:
    (a) ~~$r \leftarrow r_1 \oplus r_2$~~
    (b) $(\mathcal{D}_\mathcal{V}, \mathcal{Y}_\mathcal{V}) \leftarrow \mathsf{DataGen}(r)$
    (c) run $b \leftarrow \mathsf{ZK.Verify}(\mathsf{pp}, (\mathsf{com}, \mathcal{D}_\mathcal{V}, \mathcal{Y}_\mathcal{V}), \pi)$
    (d) locally output $(\mathcal{D}_\mathcal{V}, \mathcal{Y}_\mathcal{V}, b)$

- Adult ($d = 14$) Repository (1996): The ground truth label indicates whether an individual's income is $\geq 50,000$. Among the considered individuals, 75% have a salary below 50K. We use the gender (Male, Female) as a sensitive attribute, with 67% of the datapoints being Males.

- American Community Survey Income ($d = 10$) Ding et al. (2021): Same as in above, the ground truth label indicates whether an an individual's income is $\geq 50,000$. We use the gender (Male, Female) as a sensitive attribute, with 52% of the datapoints being Males.

We chose Default Credit and Adult since they are the two largest datasets used by C-PROFITT. As the Adult dataset is by now outdated, we also conducted the experiment using a replacement dataset suggested in Ding et al. (2021).

### G.3 ADDITIONAL PERFORMANCE AND EFFECTIVENESS EVALUATIONS

In Table 3, we present the performance of our AudiFair protocol for checking Demographic Parity/Equalized Odds/Mean Residual Difference on synthetically generated test data based on the three datasets mentioned above. For these experiments, 70% of each dataset was used to train the tree and the remaining 30% was used to train a CTGAN synthesizer. Then the synthesizer generated $n$ synthetic data points, which were used as the test dataset for checking Equalized Odds.

Table 3: Performance of our AudiFair protocol for checking Demographic Parity/Equalized Odds/Mean Residual Difference. 'SMP' stands for Security against Malicious Prover. $n$ is the size of the test dataset, $d$ is the number of features, and $h$ is the height of the decision tree. 'Data' includes both computation of $h_1, h_2$ and an execution of DataGen. The running times are in seconds and based on experiments conducted on an Amazon EC2 c7a.12xlarge instance with 96GB RAM. In these experiments, we parallelized FFT and elliptic curve operations of the underlying Groth16 NIZK Setup and Prove using 48 vCPUs.

| | $(n, d, h)$ | Setup (s) | Data (s) | Prove (s) | Verify (s) | Comm. (kB) | SMP |
|---|---|---|---|---|---|---|---|
| ACSIncome | $(15000, 10, 10)$ | 103 | 288 | 52 | $< 1$ | 0.6 | ✓ |
| Credit | $(9000, 23, 10)$ | 73 | 239 | 46 | $< 1$ | 0.6 | ✓ |
| Adult | $(13500, 14, 10)$ | 93 | 295 | 57 | $< 1$ | 0.6 | ✓ |

Figure 5: Comparison of Equalized Odds and Mean Residual Difference scores $\Delta$ calculated using the reference dataset and CTGAN-generated synthetic data with varying epochs (the number of times each data point is used during synthesizer training), with the sensitive attribute set to gender. Dashed lines represent $\Delta$ derived from the reference dataset. Both fair and biased models are trained on 70% of each reference dataset, and the remaining 30% is used for calculating $\Delta$ for the reference set. For the synthetic data, we use the same 30% of the reference dataset to generate synthetic data with CTGAN.

The results include the setup time, commitment time, data generation time, proof generation time, verification time, and communication bandwidth. The experiments were conducted on an Amazon EC2 c7a.12xlarge instance with 96GB RAM, utilizing 48 vCPUs for parallelizing FFT and elliptic curve operations during the Groth16 NIZK Setup and Prove phases. For all performance experiments, we fixed the number of epochs for the CTGAN model to 300 steps, which is the default value in the SDV library, and we conservatively set $k = 1024$ to retain 128 bits of security, assuming that the adversary's query budget to the random oracle is $2^{128}$.

In Fig. 5, we compare the Equalized Odds and Mean Residual Difference scores $\Delta$ calculated using the reference dataset and CTGAN-generated synthetic data with varying epochs, with the sensitive attribute set to gender. To provide two baselines for each dataset, we trained a "fair" model without using the sensitive attribute, and an artificially "biased" model on a manipulated dataset where the ground truths are fixed to 1 if the sensitive attribute is 1. Moreover, in Fig. 6, we compare the Demographic Parity score $\Delta$ calculated with the reference dataset and with TVAE-generated synthetic data Xu et al. (2019) with varying epochs.

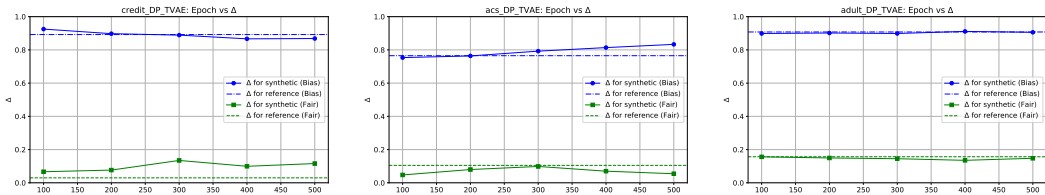

Figure 6: Comparison of Demographic Parity score $\Delta$ calculated with reference dataset and with TVAE-generated synthetic data with varying epochs. Both the fair and biased models are trained on 70% of each reference dataset. The remaining 30% is used for calculating $\Delta$ for the reference set. For the synthetic data, we use the same 30% of the reference dataset to generate synthetic data with TVAE.