# OpenReview forum: "AudiFair: Privacy-Preserving Framework for Auditing Fairness"
_ICLR.cc/2026/Conference — Submitted to ICLR 2026_

### Official Review · Reviewer_C9np · 2025-10-22

**Soundness:** 4
**Presentation:** 3
**Contribution:** 2
**Rating:** 2
**Confidence:** 3

**Summary:**

The submission describes a system for auditing/evaluating the fairness of a machine learning model such that the model itself is not leaked to the verifying party, no potentially private data is needed in the process, and neither model provider nor verifier need to be *honest*, i.e. follow the protocol. To achieve these properties, a number of cryptographic primitives are instantiated: the model provider has to provide a formal *commitment* to ensure that they actually compute the evaluation of the promised model on the provided data, not a potentially manipulated version or manipulated output. A zero-knowledge proof scheme allows the model provider to prove to the verifier that a desired fairness proper holds, without leaking any other information. Finally, agreed on synthetic data is used as input to the system to avoid privacy concerns.
Most of the manuscript is used explains these cryptographic concepts, and their application to the specific task at hand. Afterwards, results of experiments are reported for decision tree models in reasonably large/real-world datasets. One observation here is that the proposed implementation is much faster than prior work that furthermore provided weaker guarantees.

**Strengths:**

+ cryptographic safety for ML methods is an intersting and relevant topic
+ the described method makes sense for the task at hand
+ the presentation does a good job at trying to explain the method also to non-experts readers
+ the reported speedups are impressive

**Weaknesses:**

The main weakness of the manuscript is its unclear scientific contribution. In general, I could see multiple axes of potential innovation: new crypto, new machine learning, new contributions to  systems/implementation, new insights at the interface of two or more of these.

On the crypto side, the manuscript does not seem to propose new methods, but exloit on existing primitives. Essentially, it follows the common pattern of replacing a trusted party (that would get the model and the data and compute the fairness certificate) by cryptography. The fact that existing component are used is a good thing for the credibility to the actual safety of the proposed system and the possibility of practical implementation, but it does not constitute much of scientific novelty, besides proof that the combination of the component still fulfills the desired privacy criteria.
On the ML side, the manuscript also does not propose new methods. It deals with the question how to evaluate a known quantity (fairness) for a known and already trained ML model. Again, this is good, because it allows widespread applicability, but there is no scientific novelty on the ML side.
One novel aspect is on the systems side. Running crypto in practice requires a substantial amount of system design and implementational effort, e.g. to achieve efficiency. However, the manuscript is not written as a systems paper. Aspects of implementation are largely skipped, only the underlying libraries are mentioned. Even the specific choice of cryptographic compenet is not justified, except SWIFFT/Groth16/zkSNARK for trees, which was introduced in prior work Zhang et al. (2020). For example, why Groth16? That leads to small size for the proof, but in this context, efficiency seems more of the focus. Instead of the SWIFFT hash, why not e.g. Poseidon?
Ultimately, the main contribution seems to be of connecting the fields, demonstrating to the machine learning community how (existing) cryptography allows solving problems without a trusted party. I agree that the majority of the machine learning community is not aware of the possibilities of modern crypto. However, such a demonstration might be better placed in a position paper or a tutorial, not in a paper submission to the core technical track.

Besides the above, I find it a shortcoming that the procedure requires synthetic data, but the limitations of this are not discussed. Fairness is a distribution dependent quantity, and it is not discussed under which conditions, synthetic data will be a suitable proxy for real data. Furthermore, practical synthetic data generators often have subtle artifacts, which might offer a path to break the proposed protocol: if the model provider can detect such artifacts, they might commit to a model that returns fair (e.g. random) answers on synthetic data, but otherwise behaves unfairly.

**Questions:**

* I would like to understand your reasoning why you think ICLR is the right publication venue for your work. Please clarify what you consider the main scientific contribution of your work? Is it on the crypto, ML or systems side, or does it lie a new combination of existing tools, or increasing the awareness of the ML community to the existence of certain cryptographic tools? Who do you think will benefit from your work, if it is published at ICLR?

* Is the proposed scheme still secure if synthetic data is distiguishable from real data?

* Please explain your choices of SWIFFT and Groth16 zkSNARK.

---

> ### Author Response · Authors · 2025-11-26
> **Author's response to reviewer C9np (1)**
>
> ## 1. Scientific contributions and suitability to ICLR
> > Please clarify what you consider the main scientific contribution of your work? Is it on the crypto, ML or systems side, or does it lie a new combination of existing tools, or increasing the awareness of the ML community to the existence of certain cryptographic tools?
>
> Our work goes beyond a simple application of ZK proofs by introducing new scientific contributions that, for the first time, address the gap between naive ZK use and the actual goal of privacy-preserving auditing:
>
> - **Threat models and security notions for cryptographic auditing**: Prior work applied ZK proofs to auditing, _assuming standard ZK and (knowledge) soundness were sufficient_, without considering fairness-specific requirements. We identify this subtle gap and propose tailored security notions to address it:
>     - **Dishonest Provider Robustness**: Knowledge soundness of proof systems does not prevent a model provider from biasing the audit by choosing a skewed dataset (see attack on Confidential-PROFITT, Appendix D). We address this with a tailored game-based security model (Appendix F.1).
>     - **Test Data Transparency**: ZK alone does not prevent a dishonest verifier from using malicious test data to make a fair model appear unfair using a maliciously crafted test dataset (see our response to Reviewer pP8e). We formalize test data transparency to address this, as detailed in Appendix F.1.
>
> - **Formal proof of a generic framework**: We combine secure coin tossing, synthetic data generation, and ZK proofs to achieve all identified properties, with detailed hybrid arguments in Appendix F.2. To our knowledge, this is the first protocol to do so (see Table 1). Integration of secure coin tossing and synthetic data are crucial because they prevent either party from choosing adversarial test data (i.e., statement in the ZKP). While the reviewer writes that "synthetic data is used as input to the system to avoid privacy concerns," (and we agree that synthetic data may indeed help with privacy concerns), our main goal here is test data transparency, which we will clarify in the revision.
>
>
> > I would like to understand your reasoning why you think ICLR is the right publication venue for your work.
>
> Given these contributions, we believe that our work fits ICLR's CFP, which highlights "fairness, safety, privacy." While not novel in every cryptographic primitive, ICLR has previously accepted similar ML applications using existing cryptography, e.g., [Confidential-PROFITT](https://openreview.net/forum?id=iIfDQVyuFD), [CipherPrune](https://arxiv.org/pdf/2502.16782). Comparable venues also accept such work, e.g., [ExpProof, ICML '25](https://icml.cc/virtual/2025/poster/44593), [OATH, NeurIPS '25](https://arxiv.org/pdf/2410.02777?).
>
> > Who do you think will benefit from your work, if it is published at ICLR?
>
> We believe both cryptographers and ML researchers will benefit: cryptographers can use our new threat models and security notions to design better cryptographic auditing schemes for ML, while ML researchers can apply our approach to combine cryptographic tools for advanced fairness auditing. We further hope that the attack on Confidential-PROFITT (which we outline in Appendix D) will help ML researchers to better understand various cryptographic guarantees as well as limitations of prior cryptographic approaches published at ML venues.
>
> ## 2. Fairness w.r.t. synthetic vs real data
> While we are not aware of prior work on computational/statistical indistinguishability between synthetic and real data (though this could be an interesting research direction), the main goal of any synthetic data generation is to closely match the real data distribution inferred from the input training data.
> This goal has been achieved empirically for CTGAN and TVAE: they perform well by likelihood fitness, which measures the likelihood of synthetic data under the distribution from which the training data is sampled (see [Xu et al., Section 5.2](https://arxiv.org/pdf/1907.00503)).
> For fairness metrics like demographic parity, our Figures 2, 5, and 6 show similar results for real and synthetic data.
> Recent works such as [P2NIA]("https://arxiv.org/pdf/2504.00874") as well as [Yuan and Wang](https://arxiv.org/pdf/2504.21634) also confirm that synthetic data preserves key fairness properties across datasets and models.

---

> > ### Comment · Reviewer_C9np · 2025-11-28
> >
> > Thank for the clarifications. Since you didn't specifically give answers to parts of my first two questions, am I right to interpret what you wrote in the following way?
> > * for my second question, if synthetic data was distinguishable from real data, your system would indeed not be secure anymore, but you do not consider this an actual problem, because synthetic data generation is good enough.
> > * for my first question, you see your main contribution on the conceptual side (new thread model/security notions) and on the crypto side in the proof of the new protocol?
> >
> > I won't need a long answer, just let me know if I understood you correctly or, if not, what I missed.

---

> > > ### Author Response · Authors · 2025-12-01
> > > **Response to the comments**
> > >
> > > > for my first question, you see your main contribution on the conceptual side (new thread model/security notions) and on the crypto side in the proof of the new protocol?
> > >
> > > Yes, we believe the submission has conceptual contributions, including the threat model and security notions, as well as cryptographic contributions in the new protocol and formal security proofs, as the reviewer noted. In addition:
> > >
> > > - Our submission shows that prior work (Confidential-PROFITT) is insecure under our practically relevant threat model by presenting a concrete attack strategy for dishonest model providers in Appendix D.
> > >
> > > - We also contribute on the system side by implementing a proof-of-concept instantiation and demonstrating good performance, as stated in our response to Q3. In response to reviewers gZeC and XTQx, we additionally implemented the framework with neural networks and XGBoost, which we believe is a further contribution.
> > >
> > >
> > > > for my second question, if synthetic data was distinguishable from real data, your system would indeed not be secure anymore, but you do not consider this an actual problem, because synthetic data generation is good enough
> > >
> > > No, our scheme is secure, because we **do not** require full computational indistinguishability between the real distribution $D$ and the synthetic distribution $D'$. _The requirement we use is strictly weaker_: it suffices that the audited model class (e.g., decision trees) cannot distinguish synthetic from real data with high probability. This is justified, because (1) our scheme verifies (via our cryptographic zero-knowledge proof) that the provider uses the model of a given type, and (2) our scheme explicitly requires that the provider determines a model *before* observing synthetic data for auditing.
> > >
> > > Below we provide a detailed discussion on this, which we are happy to include in the paper. We will first recall our auditing scheme, and clarify the difference between computational indistinguishability in cryptography and our specific requirements.
> > >
> > > ## Recap: Our Auditing Scheme
> > > We restate the setting of our two-party protocol, in which an auditor interacts with a model provider to check fairness properties of a private classification model. Although the auditing scheme keeps the model parameters private, _it still ensures, via a zero-knowledge proof, that the provider outputs a model in a specified class $\mathcal{M}$_. For example, in our instantiation in Section 5, $\mathcal{M}$ consists of decision trees of constant depth.
> > >
> > > Assuming the standard security properties of the cryptographic building blocks stated in Theorem 1, the overall structure of our protocol is as follows:
> > >
> > > 1. Model Provider determines a model $M\in\mathcal{M}$ by outputting a cryptographic commitment to it.
> > > 2. Auditor checks fairness conditions based on the classification $M(x_i)$, where $x_i\sim D'$ is a freshly sampled synthetic data point.
> > >
> > > In particular, _an adversarial model provider does not receive a synthetic data point before committing a model_.
> > >
> > > ## Indistinguishability in Cryptography vs Our Assumption
> > > We now point out subtle differences between indistinguishability and our assumption.
> > >
> > > - **(Computational) Indistinguishability**: In cryptography, indistinguishability is a strong requirement as it needs to hold with respect to _arbitrary probabilistic polynomial time (PPT) algorithms_. Formally, two distributions $D$ and $D'$ are said to be indistinguishable if no PPT algorithm can reliably decide whether a fresh sample $x$ is drawn from $D$ or $D'$.
> > >
> > > - **Our Assumption**: We require only that no model $M$ in the class $\mathcal{M}$ (rather than an arbitrary PPT algorithm) can reliably distinguish $D$ from $D'$. This is because in our scheme, a sample is not evaluated by an arbitrary PPT algorithm; it is evaluated only by a model $M\in\mathcal{M}$.
> > >
> > > Thus, _even if there might exist a powerful PPT algorithm that distinguishes $D'$ from $D$, our scheme is secure as long as a limited class of models cannot distinguish_.
> > >
> > > Furthemore, if an auditor operates with a tolerance margin (say fairness within a small constant of the required threshold), we can further relax our requirment accordingly. Formally, unlike computational indistinguishability, which requires every PPT adversary to have _negligible_ advantage in distinguishing two distributions, we do not need such a strong requirement. In practice, auditors may tolerate slight (but non-negligible) deviations from the target threshold (say, an auditor aiming for demographic parity of 0.7 might consider 0.71 acceptable still). Thus, the advantage of a model $M \in \mathcal{M}$ in distinguishing between $D$ and $D'$ can be relaxed even further to a small constant.

---

> ### Author Response · Authors · 2025-11-26
> **Author's response to reviewer C9np (2)**
>
> ## 3. On Groth16 and SWIFFT
> Groth16 is widely used for compact, verifier-efficient proofs. We focus on settings where auditors have less capacity than model providers; in practice, model providers are well-funded companies capable of training complex ML models, whereas auditors do not provide similar services.
> While newer systems like [Polymath](https://eprint.iacr.org/2024/916) and [Pari](https://eprint.iacr.org/2024/1245) slightly reduce communication, we chose Groth16 for its maturity.
> Other options, such as VOLE-based ZK, may suit cases needing faster runtime or where interactivity/communication is less of a concern.
>
> We use SWIFFT as inherited from Zhang et al.'s ZKP for decision trees.
> While an optimized implementation, including Poseidon is a promising direction for future work, the goal of this paper is to identify security properties not covered by traditional ZK and soundness notions, prove the security of the generic construction, and demonstrate that our careful orchestration of existing primitives leads to good performance.

---

### Official Review · Reviewer_XTQx · 2025-10-26

**Soundness:** 3
**Presentation:** 3
**Contribution:** 2
**Rating:** 6
**Confidence:** 3

**Summary:**

This paper, AudiFair, presents a cryptographic framework for auditing AI model fairness while preserving the model's privacy. The main idea is to address issues with existing methods, such as model providers cheating or real-world users having to be involved in the audit process. The paper claims to be the first to combine three features: model privacy, robustness against a dishonest provider, and transparent test data generation.

The protocol works in three phases: the provider "commits" to their model, then the provider and verifier jointly generate a synthetic test dataset, and finally, the provider uses a zero-knowledge proof (ZKP) to demonstrate that their model is fair on that dataset. The authors implemented and tested this for decision trees. They report a substantial improvement in communication size over a prior work called C-PROFITT.

**Strengths:**

* The paper tackles an important practical problem: how to verify a model's fairness when the model owner cannot or will not reveal it. The goal of achieving this without involving real-world users is a good one and seems to be a key advantage over some related work.

* he paper is clearly written. The introduction does a good job of explaining the problem, and Table 1 was particularly helpful for understanding how this work claims to be different from its competitors.

* Experimental Validation: The authors use synthetic data for the audit, which could be a concern. However, they sensibly check that the fairness scores (like Demographic Parity) on the synthetic data are close to the scores on the real data, which provides some validation for this approach.

* Communication Efficiency: The reduction in communication bandwidth (the final proof size) is a notable practical improvement.

**Weaknesses:**

* Computational Cost: A major drawback is the prover's computation time. The paper states this is ~10x higher than the C-PROFITT baseline. This seems like a very high price to pay for the added security. This high overhead might make the system impractical for many real-world scenarios where audits need to be run quickly or frequently.

* Limited Model Scope: The evaluation is only for decision trees. This is a very simple model class. Most modern AI systems with serious fairness concerns use much more complex models like deep neural networks or large gradient-boosted ensembles (e.g., XGBoost). It's not at all clear how or if this framework could ever be efficient enough for those models, and the paper doesn't provide a path forward.

* Unclear DataGen Security: I was a bit confused about the security of the DataGen step. The protocol secures the random seed used to generate the data, but it seems to assume the synthetic data generator algorithm itself is honest. What stops a provider from training and committing to a generator that, even with a random seed, only produces "easy" or non-representative data, making the fairness check trivial to pass? This seems like a potential gap.

**Questions:**

* How realistic is the "future work" of applying this to models like XGBoost or neural networks? My understanding is that ZK-proving for these models is extremely expensive, so it's hard for me to see how this could be practical.

* Could you please clarify the assumption about the DataGen algorithm ? Does the protocol assume both parties have access to and agree on a trusted, public version of the generator? What happens if the provider is the one who supplies the generator?

---

> ### Author Response · Authors · 2025-11-26
> **Author's response to reviewer XTQx (1)**
>
> We thank the reviewer for acknowledging the key advantage of our submission over related work.
>
> ## 1. Extension to neural networks and XGBoost
> > Q.1 How realistic is the "future work" of applying this to models like XGBoost or neural networks? My understanding is that ZK-proving for these models is extremely expensive, so it's hard for me to see how this could be practical.
>
> > W.2 Limited Model Scope: The evaluation is only for decision trees. This is a very simple model class. Most modern AI systems with serious fairness concerns use much more complex models like deep neural networks or large gradient-boosted ensembles (e.g., XGBoost). It's not at all clear how or if this framework could ever be efficient enough for those models, and the paper doesn't provide a path forward.
>
> As our framework is modular and can be instantitated with any ZK proof of inference, we can extend our framework to XGBoost and neural networks is a realistic direction, thanks to the recent advancements in efficient ZK proofs. Since Reviewer gZeC has raised a similar concern, in the next comment, we first outline ZK Proofs of Fairness for neural networks and XGBoost, and then provide preliminary experimental results.
>
>
> ## 2. Assumption about DataGen
> > Q.2 Could you please clarify the assumption about the DataGen algorithm ? Does the protocol assume both parties have access to and agree on a trusted, public version of the generator? What happens if the provider is the one who supplies the generator?
>
> > W.3 Unclear DataGen Security: I was a bit confused about the security of the DataGen step. The protocol secures the random seed used to generate the data, but it seems to assume the synthetic data generator algorithm itself is honest. What stops a provider from training and committing to a generator that, even with a random seed, only produces "easy" or non-representative data, making the fairness check trivial to pass? This seems like a potential gap.
>
> Yes, we assume that the DataGen algorithm is honestly designed and agreed upon by both parties in advance. It can be simply any well-established public synthetic data generation algorithms such as CTGAN and TVAE, which we used in our experiments.
>
> We do not allow the provider to supply their own generator, since this would compromise robustness against dishonest providers. A malicious provider could design a synthetic generator that outputs a skewed dataset, enabling similar attacks as described for Confidential-PROFITT (see Appendix D).
>
> ## 3. On computational costs
> > W.1 Computational Cost: A major drawback is the prover's computation time. The paper states this is ~10x higher than the C-PROFITT baseline. This seems like a very high price to pay for the added security. This high overhead might make the system impractical for many real-world scenarios where audits need to be run quickly or frequently.
>
> While our prover requires more computation time, our verification is significantly cheaper due to the Groth16 proof system.  We focus on settings where auditors have less capacity than model providers; in practice, model providers are well-funded companies capable of training complex ML models, whereas auditors do not provide similar services. Our framework is not tied to any specific proof system, and the experimental results should be viewed as benchmarks for one of many possible instantiations. Groth16 could be certainly replaced with other ZK systems that balance prover and verifier complexity; for example, VOLE-based ZK (as used in CP) may be preferable when faster runtime or reduced interactivity/communication is desired.

---

> ### Author Response · Authors · 2025-11-26
> **Author's response to reviewer XTQx (2)**
>
> ### Applicability to Neural Networks
> Following our template for ZK Fairness Proof on p.6, we require:
>
> 1. ZK proof of committed model $M$.
> 2. ZK proof of $n$ inferences $\hat{\mathcal{Y}} = \\{\hat{y}_i\\}_i$, where $\hat{y}_i = M(a_i)$ for a dataset $\mathcal{D} = \\{a_i\\}_i$.
> 3. ZK proof of fairness constraints: $\mathsf{FM}(\hat{\mathcal{Y}}, \mathcal{Y}, \mathcal{D}, s) < t$.
>
> We outline a general approach to instantiating Steps 1 and 2 above for neural networks:
>
> - **Committed Inputs**: The first critical step is for the Prover to commit to the private model weights $W$, allowing the Verifier to receive _masked versions of the weights_, typically denoted by $[W]$. Similarly, the Prover commits to all internal values that arise during inference $y = M(a)$. The remaining goal of the ZK proof is then to validate that all internal values satisfy the constraints defined by each layer.
> - **Basic Addition and Multiplication**: Typical ZK proofs for arithmetic circuits natively support validation of addition (given $[x], [y], [z]$, check $z = x + y$) and multiplication (given $[x], [y], [z]$, check $z = x \cdot y$) constraints. More complex constraints can be checked by composing these atomic subprotocols.
> - **Linear Layers**: The essential operation in linear layers is matrix-vector multiplication $W \cdot x$, which can be expressed as a combination of basic addition and multiplication operations.
> - **Non-Linear Layers:** As summarized by [Hao et al.](https://eprint.iacr.org/2025/507), ZK subprotocols for common non-linear functions in machine learning, such as ReLU, Softmax, Sigmoid, and GELU, are well-studied. For example:
>     - ZK proof for ReLU $y=\max(0,x)$ can be implemented by computing a bit $b \gets (x \geq 0)$ and checking $y = b \cdot x$. This is achieved using standard ZK building blocks for comparison and multiplication.
>     - ZK proof for Sigmoid $p = \frac{1}{1+e^{-x}}$, which can be rewritten as $p=\frac{e^{-|x|}}{1+e^{-|x|}}$ if $x<0$, and thus can be implemented using ZK subprotocols for comparison, exponential, and division. ZK proof of exponentiation $e^x$ with $k$-bit $x$ is realized by decomposing $x$ into bits $x_1, \ldots, x_k$, computing bitwise exponentiation $e^{x_i}$ using a look-up argument, and then aggregating the scaled $e^{x_i}$ multiplicatively. The division protocol is also  detailed in Hao et al.
>
>
> To demonstrate applicability of our approach, we have implemented our ZK Fairness Proof for the feed-forward neural network (FFNN), and obtained the following result on an AWS EC2 g5.8xlarge instance:
>
> | \# Samples | \# Features | \# Parameters | \# Layers | Prover Runtime (min) | Verifier  Runtime (min) |
> | --------   | ----------- | ------------- | --------- | -------------        | ------- |
> | 5000       |  20         |  485          | 3         | 6.8                | <1 |
>
> We can provide further benchmarks for larger models. Overall, as in our setting the audit needs to be done only once with no time constraint, we believe the approach is entirely feasible.
>
>
>
> ### Applicability to XGBoost
> Similar to neural networks, following our template for ZK Fairness Proof on p.6, we can adapt our framework to an XGBoost model. Recall that XGBoost prediction proceeds as follows:
>   1. **Tree Evaluation**: evaluate each tree $T_k$ on input $x$ to obtain a weight $w_k$ for $k=1,\ldots,K$,
>   2. **Score Aggregation**: compute an aggregated score $z = \sum_{k} \eta\cdot w_k$, and
>   3. **Probability Transformation**: apply the sigmoid to compute probability $p = \frac{1}{1+e^{-x}}$.
>
> Step 1. is essentially the same as evaluation of $K$ decision trees and is the most expensive, as its cost scales with $K$ and tree depth $h$. The cost of ZK proofs for Steps 2. and 3. is negligible; 2. is a simple linear operation that can be efficiently computed in ZK, and 3. can be realized using standard ZK building blocks for comparison, exponentiation, and division (see, e.g., [Hao et al., Section 6](https://eprint.iacr.org/2025/507.pdf#section.6)).
>
> To demonstrate applicability of our approach, we have implemented our ZK Fairness Proof for XGBoost, and obtained the following result on an AWS EC2 g5.8xlarge instance.
>
> | \# Samples | \# Features | \# Trees | Depth | Prover Runtime (min) | Verifier  Runtime (min) |
> | --------   | ----------- | ------------- | --------- | -------------        | ------- |
> | 5000       |  20         |  50          | 5         | 3.5                 | <1 |

---

### Official Review · Reviewer_pP8e · 2025-11-01

**Soundness:** 2
**Presentation:** 2
**Contribution:** 2
**Rating:** 2
**Confidence:** 3

**Summary:**

The paper proposes defense against malicious auditors when verifying fairness of a model in a privacy-preserving fashion. It claims to do so by having the prover and verifier agree on a randomness, which is used to generate synthetic data for fairness auditing.

**Strengths:**

The idea that there needs to be protection against malicious verifiers seems interesting.

**Weaknesses:**

1. I think exactly how the verifier can manipulate fairness scores and therefore the corresponding threat model the algorithm is trying to protect against is not clear. As far as I can understand, it could be because verifier gives model A an easy audit dataset vs. model B. Authors claim the issue could be prevented for giving a harder dataset to B (since in the protocol both verifier and prover B in this case will agree to a randomness). However, then wouldn't everyone want an easier audit dataset to pass? I think the point should be to come up with a scheme that leads to true audits. To see exactly what your defense protects against, please add more clarity in the threat model.

2. What does it mean to "fail the audit despite holding a fair model."?? -- all auditing datasets are valid, some can successfully find the flaws in the model others cannot. I am failing to understand how the model not succeeding on an audit dataset is a problem -- it just means the audit dataset did its job.

3. The abstract is really badly written. Major part of the abstract should talk about your contribution. From the abstract it seems like this is the first paper proposing ZKPs for fairness audits. Fig1 also completely misses the point of the paper. The focus of the paper is verifier, not prover. Preventing against dishonest provers has been seen in the literature.

4. How do you get lower numbers than CP? I couldn't understand your contributions which reduced the time overhead.

**Questions:**

See above

---

> ### Author Response · Authors · 2025-11-26
> **Author's response to reviewer pP8e (1)**
>
> We thank the reviewer for acknowledging the importance of protection against malicious verifiers.
>
> > 1. I think exactly how the verifier can manipulate fairness scores and therefore the corresponding threat model the algorithm is trying to protect against is not clear. As far as I can understand, it could be because verifier gives model A an easy audit dataset vs. model B. Authors claim the issue could be prevented for giving a harder dataset to B (since in the protocol both verifier and prover B in this case will agree to a randomness). However, then wouldn't everyone want an easier audit dataset to pass? I think the point should be to come up with a scheme that leads to true audits. To see exactly what your defense protects against, please add more clarity in the threat model.
>
> ### Our threat model
> We believe there may be a misunderstanding. We do not claim to give a harder dataset to the model provider. Instead, our solution addresses the following threat models simultaneously by requiring both parties to contribute to the test dataset using a pre-agreed synthetic data generation process and secure coin tossing, followed by a custom zero-knowledge proof:
>
> 1. A malicious model provider who attempts to pass the audit with an unfair model.
> 2. A malicious verifier who attempts to make an honest model provider fail the audit, even if the model satisfies fairness in the real world.
>
> Protocols that use a dataset chosen by the provider, such as Confidential-Proffit (CP), are vulnerable to (1). A naive countermeasure is to let the verifier choose the dataset, which in turn makes the scheme vulnerable to (2) (see the next paragraph). This is why we designed the protocol that carefully combines commitment, randomness agreement for synthetic data generation, and zero-knowledge proof, so that neither party can bias the auditing outcome maliciously.
>
>
> ### How malicious verifier can manipulate the fairness score
> To see how a malicious verifier can make a fair model appear unfair with respect to a carefully crafted dataset, consider the following toy example: a binary classification model $M(x)$ that outputs 1 if and only if $x.{}a=1$, where the feature $a$'s distribution in real world is independent of the sensitive attribute $x.{}s$. For instance, if $M$ decides whether an applicant $x$ is approved for credit loan, one could consider the sensitive attribute $s$ as 'gender' and $a$ as the first digit of zip code, which are unlikely correlated.
> It is easy to see that such a model exhibits demographic parity $=$ 0 (i.e., perfectly fair) if $x$ is sampled from a real world population distribution, because we have that $\Pr[M(x)= 1 \,|\, x.s=1]=\Pr[x.a=1 \,|\, x.s=1]=\Pr[x.a=1]=\Pr[x.a=1 \,|\, x.s=0]=\Pr[M(x)= 1 \,|\, x.s=0]$.
>
> However, if a malicious verifier can send a skewed dataset where the feature $a$ is highly correlated with $s$, e.g., a dataset in which male applicants tend to have the zip code starting with '1', then the model $M$ exhibits a large demographic parity with such a test dataset. The issue here is that in general, there is no guarantee that an adversarially crafted test dataset is identically distributed as the training dataset. In this case, a malicious verifier could make the provider fail the audit even if the model behaves fairly in real-life conditions.
>
>
> In fact, this is exactly why Confidential-PROFITT opts for a solution without verifier-supplied datasets, as they remark on page 1:
>
> > This may lead to a form of unhelpful interaction between the company and the auditor, in which the company could deny a model is unfair by claiming that the reference dataset does not belong to the training distribution used, or the auditor can forge a reference dataset that could be used to blame the company for unfair predictions.
>
> Thus, our framework addresses both of these concerns while simultaneously protecting against dishonest provers, overcoming the limitation of CP (see Table 1).
>
>
> > 2. What does it mean to "fail the audit despite holding a fair model."?? -- all auditing datasets are valid, some can successfully find the flaws in the model others cannot. I am failing to understand how the model not succeeding on an audit dataset is a problem -- it just means the audit dataset did its job.
>
> Please refer to our response to Question 1. If the verifier behaves maliciously, we cannot assume that all auditing datasets are valid, as they may not represent the distribution for which the model is meant to be used.

---

> > ### Author Response · Authors · 2025-11-26
> > **Author's response to reviewer pP8e (2)**
> >
> > > 3. The abstract is really badly written. Major part of the abstract should talk about your contribution. From the abstract it seems like this is the first paper proposing ZKPs for fairness audits. Fig1 also completely misses the point of the paper. The focus of the paper is verifier, not prover. Preventing against dishonest provers has been seen in the literature.
> >
> > While we are happy to update Fig. 1 to incorporate malicious verifiers, we believe there is a misunderstanding. While we (to the best of our knowledge) are indeed the first to provide test-data transparency, protection against fully malicious provers is equally important, and our protocol provides it as well.
> >
> > Prior work does not yet offer a complete solution for malicious provers. Some schemes targeting malicious provers, such as CP, become insecure if the prover arbitrarily modifies the training data--they explicitly assume this cannot happen, whereas we make no such restriction. Others, such as OATH, protect against fully malicious provers, but require model users to aid the auditor. Our work allows to verify fairness without requiring in-deployment audits. We will elaborate on these differences in the paper.
> >
> >
> > > 4. How do you get lower numbers than CP? I couldn't understand your contributions which reduced the time overhead.
> >
> > We achieve lower communication and verification time because we instantiate the protocol with a more compact proof system, Groth16. As we mention in Appendix E.2, Groth16 is extremely communication efficient as it only requires three group elements as a proof string _regardless of the circuit size_. In contrast, [the underlying ZK proof of CP](https://eprint.iacr.org/2020/925.pdf) is a Vector Oblivious Evaluation (VOLE)-based proof system, whose communication is linear in the circuit size. While we have an additional coin tossing phase, this incurs very little communication overhead because it only requires parties to exchange cryptographic hashes (256 bytes) and randomness for the synthetic generator. We will clarify this in the revised manuscript.

---

### Official Review · Reviewer_gZeC · 2025-11-03

**Soundness:** 3
**Presentation:** 3
**Contribution:** 2
**Rating:** 4
**Confidence:** 3

**Summary:**

This paper presents a privacy-preserving framework for evaluating fairness in machine learning models. The proposed solution leverages zero-knowledge proofs to maintain model privacy, ensure robustness against dishonest providers, and preserve the transparency of test data while evaluating fairness. The authors validate their approach using a decision tree model on the ACS dataset and compare its performance with C-PROFITT. Experimental results indicate that the proposed method achieves higher efficiency than the evaluated baselines.

**Strengths:**

1.	Developing privacy-preserving frameworks is an important and meaningful research direction for the machine learning community.
2.	The reported results appear to outperform the evaluated baselines, and the authors support their claims with empirical evidence rather than purely theoretical analysis.
3.	The paper is clearly written and well-structured, making it easy to follow.

**Weaknesses:**

1.	Similar approaches have been introduced in prior work, making the added value and contribution of the proposed framework unclear.
2.	The applicability of the proposed method is demonstrated solely using a decision tree model, which limits the scope of the evaluation.

**Questions:**

[1] While the proposed method is interesting, its novelty relative to prior work remains unclear - particularly in comparison to [1], which also defines fairness using commitment schemes, zero-knowledge proofs, and data augmentation techniques. The authors are encouraged to elaborate on how their approach differs from this prior work and what unique contributions it provides.

[2] Although I am not a cryptography expert, I recognize that constructing circuits for machine learning models, especially neural networks, is a non-trivial task. Could the authors provide more details on this aspect? Doing so would help clarify the practicality and potential applicability of the proposed framework for readers.


[1] Segal, Shahar, et al. "Fairness in the eyes of the data: Certifying machine-learning models." Proceedings of the 2021 AAAI/ACM Conference on AI, Ethics, and Society. 2021.

---

> ### Author Response · Authors · 2025-11-26
> **Author's response to reviewer gZeC (1)**
>
> ## 1. Comparison with Segal, Shahar, et al.
>
> > [1] While the proposed method is interesting, its novelty relative to prior work remains unclear - particularly in comparison to [1], which also defines fairness using commitment schemes, zero-knowledge proofs, and data augmentation techniques. The authors are encouraged to elaborate on how their approach differs from this prior work and what unique contributions it provides.
>
> We thank the reviewer for the helpful pointer. Our work differs from Segal et al. in three dimensions:
>
> **1. Security properties**
> We provide a superset of the security guarantees achieved by Segal et al.: both approaches ensure model privacy and robustness against dishonest provers, but _ours is the only one that additionally guarantees test-data transparency_, a new notion which additionally captures security against dishonest verifiers and which we formalize in Appendix F.1. We believe this notion may be of independent interest.
>
> More concretely, in the augmented-data approach of Segal et al., the verifier is responsible for locally generating the augmented data. A malicious verifier can therefore arbitrarily modify this data—or even replace it entirely—without detection, and thus manipulate the fairness score to discriminate certain model providers (see our response to Reviewer pP8e).
>
>
> **2. Protocol design.**
> The two works also differ methodologically. Segal et al. use general-purpose secure multi-party computation (MPC), while we design a custom framework incorporating commitments, secure coin tossing, and zero knowledge proofs. This is important for protocol efficiency, especially in the context of achieving test-data transparency. While their approach could potentially be augmented with our techniques to get test-data transparency, doing so within general-purpose MPC would make the construction prohibitively expensive.
>
> **3. Implementation**
> While we implemented the whole framework including a communication-efficient ZK proof and a secure tossing protocol, they do not provide an implementation or benchmarks with MPC (denoted by $\mathcal{F}_{SC}$ in the paper) instantiated for concrete models.
>
> We will include a detailed comparison in our paper.
> ## 2. Applicability and generality of our approach
> > [2] Although I am not a cryptography expert, I recognize that constructing circuits for machine learning models, especially neural networks, is a non-trivial task. Could the authors provide more details on this aspect? Doing so would help clarify the practicality and potential applicability of the proposed framework for readers.
>
> Our approach is modular and applicable to models beyond decision trees. Following our template for ZK Fairness Proof on p.6, we require:
>
> 1. ZK proof of committed model $M$.
> 2. ZK proof of $n$ inferences $\hat{\mathcal{Y}} = \\{ \hat{y}_i \\}_i$, where $\hat{y}_i = M(a_i)$ for a dataset $\mathcal{D} = \\{ a_i \\}_i$.
> 3. ZK proof of fairness constraints: $\mathsf{FM}(\hat{\mathcal{Y}}, \mathcal{Y}, \mathcal{D}, s) < t$.
>
> Steps 1 and 2 are orthogonal to fairness and well studied in the literature, and we can incorporate our additional fairness constraints (discussed in Appendix E.2) into any ZKP of inferences for any model. We discuss applicability to two examples below: neural networks and XGBoost.

---

> ### Author Response · Authors · 2025-11-26
> **Author's response to reviewer gZeC (2)**
>
> ### Applicability to Neural Networks
> We outline a general approach to instantiating Steps 1 and 2 above for neural networks:
>
> - **Committed Inputs**: The first critical step is for the Prover to commit to the private model weights $W$, allowing the Verifier to receive _masked versions of the weights_, typically denoted by $[W]$. Similarly, the Prover commits to all internal values that arise during inference $y = M(a)$. The remaining goal of the ZK proof is then to validate that all internal values satisfy the constraints defined by each layer.
> - **Basic Addition and Multiplication**: Typical ZK proofs for arithmetic circuits natively support validation of addition (given $[x], [y], [z]$, check $z = x + y$) and multiplication (given $[x], [y], [z]$, check $z = x \cdot y$) constraints. More complex constraints can be checked by composing these atomic subprotocols.
> - **Linear Layers**: The essential operation in linear layers is matrix-vector multiplication $W \cdot x$, which can be expressed as a combination of basic addition and multiplication operations.
> - **Non-Linear Layers:** As summarized by [Hao et al.](https://eprint.iacr.org/2025/507), ZK subprotocols for common non-linear functions in machine learning, such as ReLU, Softmax, Sigmoid, and GELU, are well-studied. For example:
>     - ZK proof for ReLU $y=\max(0,x)$ can be implemented by computing a bit $b \gets (x \geq 0)$ and checking $y = b \cdot x$. This is achieved using standard ZK building blocks for comparison and multiplication.
>     - ZK proof for Sigmoid $p = \frac{1}{1+e^{-x}}$, which can be rewritten as $p=\frac{e^{-|x|}}{1+e^{-|x|}}$ if $x<0$, and thus can be implemented using ZK subprotocols for comparison, exponential, and division. ZK proof of exponentiation $e^x$ with $k$-bit $x$ is realized by decomposing $x$ into bits $x_1, \ldots, x_k$, computing bitwise exponentiation $e^{x_i}$ using a look-up argument, and then aggregating the scaled $e^{x_i}$ multiplicatively. The division protocol is also  detailed in Hao et al.
>
>
> To demonstrate applicability of our approach, we have implemented our ZK Fairness Proof for the feed-forward neural network (FFNN), and obtained the following result on an AWS EC2 g5.8xlarge instance:
>
> | \# Samples | \# Features | \# Parameters | \# Layers | Prover Runtime (min) | Verifier  Runtime (min) |
> | --------   | ----------- | ------------- | --------- | -------------        | ------- |
> | 5000       |  20         |  485          | 3         | 6.8                | <1 |
>
> We can provide further benchmarks for larger models. Overall, as in our setting the audit needs to be done only once with no time constraint, we believe the approach is entirely feasible.
>
>
>
> ### Applicability to XGBoost
> Similar to neural networks, following our template for ZK Fairness Proof on p.6, we can adapt our framework to an XGBoost model. Recall that XGBoost prediction proceeds as follows:
>   1. **Tree Evaluation**: evaluate each tree $T_k$ on input $x$ to obtain a weight $w_k$ for $k=1,\ldots,K$,
>   2. **Score Aggregation**: compute an aggregated score $z = \sum_{k} \eta\cdot w_k$, and
>   3. **Probability Transformation**: apply the sigmoid to compute probability $p = \frac{1}{1+e^{-x}}$.
>
> Step 1. is essentially the same as evaluation of $K$ decision trees and is the most expensive, as its cost scales with $K$ and tree depth $h$. The cost of ZK proofs for Steps 2. and 3. is negligible; 2. is a simple linear operation that can be efficiently computed in ZK, and 3. can be realized using standard ZK building blocks for comparison, exponentiation, and division (see, e.g., [Hao et al., Section 6](https://eprint.iacr.org/2025/507.pdf#section.6)).
>
> To demonstrate applicability of our approach, we have implemented our ZK Fairness Proof for XGBoost, and obtained the following result on an AWS EC2 g5.8xlarge instance.
>
> | \# Samples | \# Features | \# Trees | Depth | Prover Runtime (min) | Verifier  Runtime (min) |
> | --------   | ----------- | ------------- | --------- | -------------        | ------- |
> | 5000       |  20         |  50          | 5         | 3.5                 | <1 |

---

### Author Response · Authors · 2025-12-03
**Discussion Summary**

We thank the reviewers for their insightful comments and for acknowledging the strengths of our submission:

- Importance of privacy-preserving fairness auditing and practical relevance of our problem
- Clear presentation, which is particularly valuable to non-experts in cryptography, and helpful comparisons to prior work (e.g., Table 1)
- Introduction of new, relevant security properties that have been missing in prior cryptographic auditing schemes (in particular, formalization of security properties against malicious verifiers)
- Empirical demonstration of performance improvements both in terms of communication and verification runtime
- Empirical validation showing that fairness with respect to synthetic data is consistent with fairness on the real data

During the discussion period, we presented the following additional results to demonstrate applicability and generality of our proposal beyond decision trees. We believe that these results will further substantiate our contributions:

- With additional experiments, we presented instantiations of the proposed framework for auditing **XGBoost** and **neural network** models. In both cases, the empirical performance indicates that such instantiations are indeed feasible in practice.

Finally, we are grateful for the reviewers’ suggestions regarding the comparison to prior work, clarifications regarding our threat model/security guarantees, and the assumptions related to synthetic data. These comments and discussion have been very helpful and fruitful, and we are happy to incorporate the resulting clarifications (along with the other points) into the paper.

---

### Meta-Review · Area_Chair_CkQg · 2026-01-05

**Summary:**

This paper presents a privacy-preserving framework to evaluate the fairness in machine learning models. The authors use the zero-knowledge proofs to maintain model privacy and ensure robustness against dishonest providers. The empirical studies on decision tree model show the proposed method has better performance than baselines. The reviewers also have concerns on the novelty and presentation of this paper.  I think this submission cannot be accepted currently.

**Reviewer Concerns:**

The reviewers have the following main concerns
1. The applicability of the proposed method is limited to decision tree.
2. The presentation should be improved.
3. The computation time of the prover is expensive.
4. The novelty looks not significant.

For the applicability, the authors include additional experiments on XGBoost and neural network. However, the other points seem to have not been addressed.

**Reviewer Scores:**

I think the reviewers will keep their scores.

---

### Decision · Program_Chairs · 2026-01-26

Reject